# Off-the-shelf proximity biotinylation for interaction proteomics

Irene Santos-Barriopedro[1,2], Guido van Mierlo[1,2,3 ✉] & Michiel Vermeulen [1,3 ✉]

Proximity biotinylation workflows typically require CRISPR-based genetic manipulation of target cells. To overcome this bottleneck, we fused the TurboID proximity biotinylation enzyme to Protein A. Upon target cell permeabilization, the ProtA-Turbo enzyme can be targeted to proteins or post-translational modifications of interest using bait-specific antibodies. Addition of biotin then triggers bait-proximal protein biotinylation. Biotinylated proteins can subsequently be enriched from crude lysates and identified by mass spectrometry. We demonstrate this workflow by targeting Emerin, H3K9me3 and BRG1. Amongst the main findings, our experiments reveal that the essential protein FLYWCH1 interacts with a subset of H3K9me3-marked (peri)centromeres in human cells. The ProtA-Turbo enzyme represents an off-the-shelf proximity biotinylation enzyme that facilitates proximity biotinylation experiments in primary cells and can be used to understand how proteins cooperate in vivo and how this contributes to cellular homeostasis and disease.

---

[1] Department of Molecular Biology, Faculty of Science, Radboud Institute for Molecular Life Sciences, Oncode Institute, Radboud University Nijmegen, Nijmegen, The Netherlands. [2] These authors contributed equally: Irene Santos-Barriopedro, Guido van Mierlo. [3] These authors jointly supervised this work: Guido van Mierlo, Michiel Vermeulen. ✉email: guido.vanmierlo@epfl.ch; michiel.vermeulen@science.ru.nl

Proximity biotinylation recently emerged as a powerful interaction proteomics technology that can be used to identify direct and indirect interactions between proteins in vivo[1–4]. This technology typically involves fusing a proximity biotinylation enzyme to target proteins of interest using CRISPR-based knock-in strategies or plasmid-based expression. Upon the addition of exogenous biotin, proteins that are in close proximity to the bait protein during the biotin pulse become biotinylated. These biotinylated proteins can subsequently be enriched from crude cell lysates using streptavidin-conjugated beads and analyzed by quantitative mass spectrometry. Various proximity biotinylation enzymes have been described, including BioID[3], BioID2[5], APEX[6], and TurboID[7]. TurboID in particular is a very attractive proximity biotinylation enzyme since it is a very fast enzyme, which labels bait-proximal proteins in minutes. Furthermore, unlike the APEX enzyme, which relies on $H_2O_2$ for its enzymatic activity, TurboID-based proximity biotinylation only requires exogenous addition of biotin to target cells and is therefore not toxic for target cells.

Proximity biotinylation enzymes have been used for various biological questions, for example for temporal profiling of DNA damage response pathways, to decipher cellular signaling pathways and for organelle-specific proteome profiling in cell culture cells and model organisms[4,8–10]. However, as mentioned above, these approaches rely on CRISPR-based knock-in or plasmid-based expression approaches to introduce a biotinylation enzyme fused to a bait protein in target cells of interest. This is not only labor-intensive but also restricts proximity biotinylation technology to cells that can be genetically engineered and maintained and propagated for a long period of time in vitro. There is therefore a need for technology to overcome this bottleneck and that facilitates proximity biotinylation workflows in primary cells in the absence of genetic engineering or transfection.

Here we present a recombinant proximity biotinylation enzyme, called ProtA-Turbo, which consists of Protein A fused to the TurboID proximity biotinylation enzyme. Upon target cell permeabilization using either fixed or non-fixed mammalian cells, the ProtA-Turbo enzyme can be targeted to baits of interest using antibodies against endogenous proteins or protein modifications. Bait proximal proteins are subsequently biotinylated upon the addition of exogenous biotin. Cells are then lysed using high stringency lysis and biotinylated proteins are affinity enriched using streptavidin-conjugated beads using appropriate negative controls. Affinity enrichments are performed in triplicate to allow a robust statistical analysis. To benchmark this method, we combined the ProtA-Turbo enzyme with antibodies against various well-characterized baits: Emerin, which resides in the nuclear envelope, the heterochromatin modification H3K9me3 and the chromatin remodeler protein BRG1, which is part of the Swi/Snf complex, in various cell types. For all these baits, confocal microscopy revealed that the ProtA-Turbo enzyme and associated biotinylation are targeted to appropriate regions in mammalian nuclei. Affinity purifications and label-free quantitative mass spectrometry revealed numerous positive controls as well as previously unreported proximal proteins for all the used baits. Finally, follow-up experiments revealed that FLYWCH1 is an H3K9me3-associated protein that interacts with H3K9me3-marked centromeric heterochromatin. In summary, the recombinant ProtA-Turbo enzyme represents an 'off the shelf' proximity biotinylation enzyme that can be used for interaction proteomics studies in fixed and non-fixed primary cells or clinical samples.

## Results
### Off the shelf proximity labeling in fixed cells using ProtA-Turbo enzyme.
With the aim to design an 'off the shelf' proximity biotinylation enzyme that does not require genetic manipulation or transfection of target cells, we generated a construct consisting of Protein A fused to the recently developed proximity biotinylation enzyme TurboID[7] (from now on referred to as ProtA-Turbo). The TurboID enzyme is a much faster, modified version of the BioID proximity biotinylation enzyme, which can biotinylate proximal proteins inside cells in minutes. This ProtA-Turbo fusion protein was expressed and purified as a His-tagged fusion protein from bacteria (Supplementary Fig. 1a). Initial activity tests revealed that the purified ProtA-Turbo enzyme efficiently triggers protein biotinylation in bacterial cells in vivo (Supplementary Fig. 1b). Furthermore, the ProtA-Turbo enzyme also triggers protein biotinylation when added to mammalian cell extracts (Supplementary Fig. 1c). We first developed a workflow for the ProtA-Turbo enzyme in fixed cells (Fig. 1a). After formaldehyde-based fixation, cells are permeabilized followed by subsequent addition of a primary antibody and the ProtA-Turbo enzyme. An isotype control IgG antibody is used as a negative control. Following several wash steps to remove unbound antibodies and enzymes, the addition of exogenous biotin triggers bait-proximal protein biotinylation. These biotinylated proteins can subsequently be enriched from crude cell lysates using streptavidin-based affinity enrichment and identified using quantitative LC–MS.

To benchmark this method, we made use of antibodies against various well-characterized nuclear baits: Emerin, BRG1, and a histone modification, H3K9me3. Immunofluorescence-based analysis revealed that ProtA-Turbo-mediated biotinylation using these antibodies is induced in a pattern that is in agreement with the known nuclear localization of the respective baits. For example, the Emerin antibody, which resides in the nuclear envelope, triggers ProtA-Turbo-mediated biotinylation in the nuclear rim, whereas the H3K9me3 antibody results in a punctuated biotinylation signal in the nucleus that is reminiscent of the DAPI-dense chromocenters in mammalian nuclei, which are enriched for H3K9me3. In contrast, the BRG1 antibody, which targets the large multi-subunit Swi/Snf complex, results in a more diffuse nuclear staining (Fig. 1b). Streptavidin-based affinity enrichment of ProtA-Turbo targeted cells with these antibodies revealed efficient enrichment of the targeted baits and associated protein biotinylation (Fig. 1c). Next, we performed ProtA-Turbo experiments followed by streptavidin-based affinity enrichment in triplicate (with each replicate starting from an individual cell suspension) with the respective baits and negative IgG controls followed by quantitative mass spectrometry analyses. As shown in Fig. 1d and Supplementary Fig. 1e, each of the three bait antibodies in combination with the ProtA-Turbo enzyme resulted in a cluster of specifically enriched biotinylated proteins, including many positive control proteins. For example, various nuclear lamina-associated proteins are enriched in the Emerin ProtA-turbo experiments, including TMPO, SUN2, and LMNB1/2. As expected, the BRG1 antibody in combination with the ProtA-Turbo enzyme-induced enrichment of numerous Swi/Snf complex subunits such as ARID1A, BRD7 and SMARCC1. Finally, the H3K9me3 antibody resulted in specific enrichment of the H3K9 methyltransferases EHMT1/2 and Suv39H1, various known H3K9me3 reader proteins (i.e. CBX5, CBX1 and UHRF1), as well as centromere-associated proteins (INCENP, CDCA8). Centromeres are known to be enriched for the H3K9me3 modification. GO term enrichment analyses for the different affinity purifications are in agreement with this: membrane and nuclear pore-associated GO terms are enriched in the Emerin experiment, Swi/snf subunits in the BRG1-ProtA-Turbo enrichment and heterochromatin and condensed chromosome terms are enriched in the H3K9me3 experiment (Supplementary Fig. 1d). As a control experiment, we generated a knock-in cell line in which we fused Emerin to TurboID. Immunofluorescence experiments revealed biotinylation signals in this cell line at the nuclear rim (Supplementary Fig. 1f), similar to

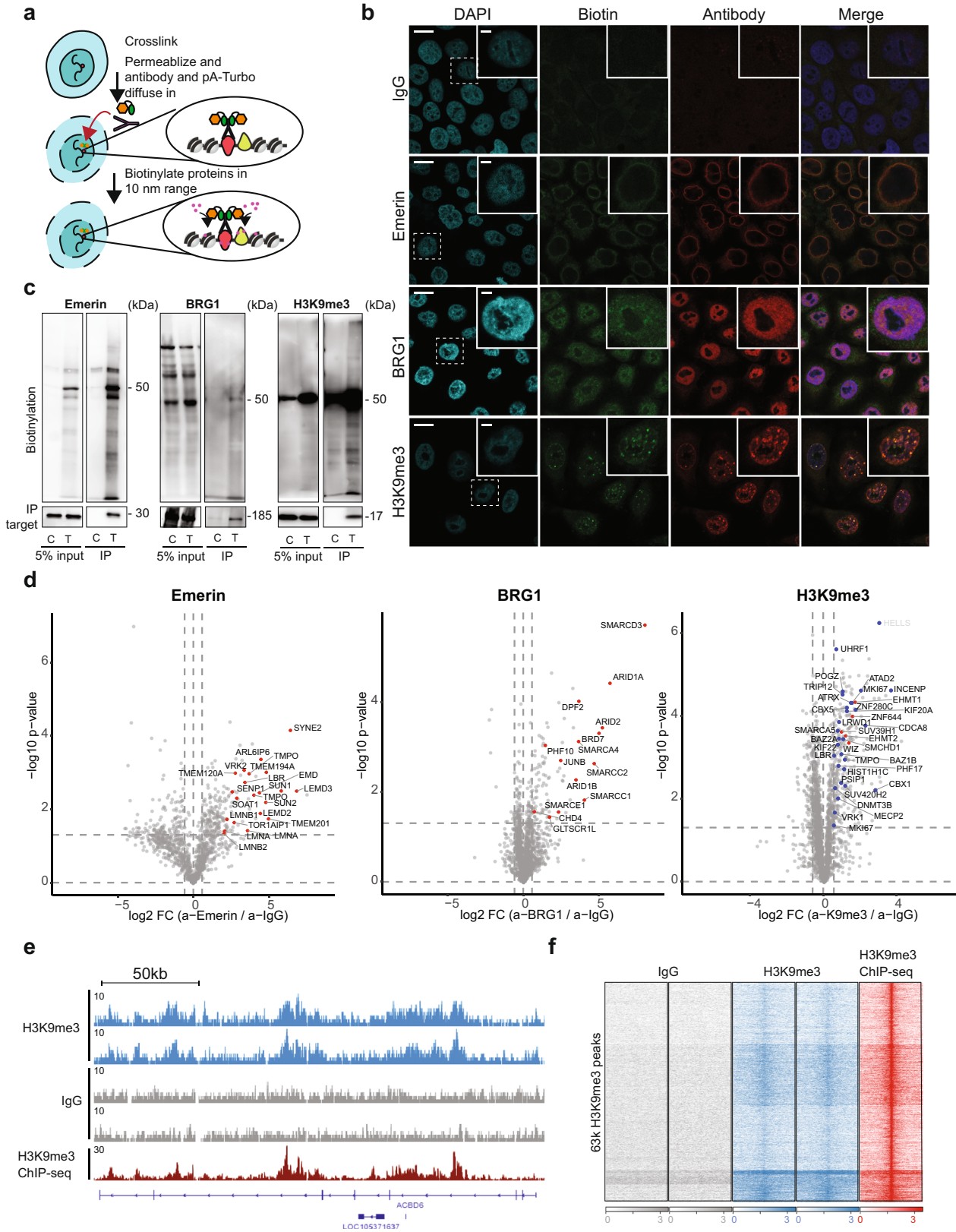

what we observed when targeting ProtA-Turbo with an Emerin antibody. Furthermore, biotin purifications of crude lysates from the Turbo-Emerin knock-in cells revealed specific enrichment of many known nuclear lamina-associated proteins, many of which were also retrieved in the protA-Turbo Emerin targeting experiment (Supplementary Fig. 1g–i). GO term enrichment analysis of the Emerin

proximity labeling experiments performed in this study are also in excellent agreement with previous interaction proteomics experiments targeting the nuclear lamina (Supplementary Fig. 1j).

To further illustrate the ability of our method to trigger antibody-dependent, bait-specific biotinylation, we performed biotin ChIP-seq using streptavidin-conjugated beads after

**Fig. 1 A ProteinA-TurboID fusion protein allows enrichment of proteins localized to a specific sub-nuclear compartment in cross-linked cells. a** Schematic outline of the method. **b** Immunofluorescence images of targeting the nuclear lamina (via Emerin), the SWI/SNF protein complex (via BRG1) or (pericentromeric) heterochromatin (via H3K9me3) in crosslinked HeLa cells. IgG was used as a control antibody. Biotinylation (in green) overlapping the antibody signal (in red) illustrates correct localization of the ProtA-Turbo fusion protein. Scale bars represent 10 μm (large panels) or 3 μm (small panels). **c** Immunoprecipitation of biotinylated proteins after targeting (T) proteins as in (**b**). IgG was used as a control (C). **d** Volcano plot of mass spectrometry analyses of biotin IPs as in (**c**). A selection of proteins known to localize to the targeted proteins are highlighted. In the H3K9me3 volcano plot, red dots indicate writer/writer complexes and blue indicates known pericentromeric proteins. Protein names in white indicate potential streptavidin contaminants (see also Source Data). x-axis represent the log2 fold change (FC) and the y-axis the −log10 p-value of comparing three IgG replicates with three specific antibody replicates. **e, f** Biotin ChIP-seq after H3K9me3 or IgG targeting with ProtA-Turbo. Number of independent experiments performed: **b** 5 (IgG), 5 (Emerin), 2 (BRG1), 5 (H3K9me3); **c** 3 (Emerin), 2 (BRG1), 5 (H3K9me3); **d** 4 (Emerin), 1 (BRG1), 3 (H3K9me3), n = 3 replicates per independent experiment; **e** 2. Source data are provided as a Source Data file.

H3K9me3 or IgG targeting with ProtA-Turbo. This experiment revealed biotinylated chromatin patterns that are significantly overlapping with previously generated H3K9me3 ChIP-seq data in the same target cells (Fig. 1e, f). Furthermore, a comparison of our ProtA-Turbo–H3K9me3 interactome with a recently published proteomics method called ChromID[11], in which proximity biotinylation is targeted to chromatin modifications using modification-specific reader domains fused to a proximity biotinylation enzyme, revealed that many H3K9me3-proximal proteins are identified using both methodologies (Supplementary Fig. 1k).

In summary, these experiments clearly validate and illustrate the value of the ProtA-turbo enzyme as an off-the-shelf proximity biotinylation enzyme in the absence of genetic manipulation or transfection of target cells, at least in a workflow with fixed cells.

**Off the shelf proximity labeling in non-fixed cells using ProtA-Turbo.** Next, to further expand the applicability of the ProtA-Turbo enzyme, we set out to establish a workflow in which the ProtA-Turbo enzyme is targeted to baits of interest in non-fixed cells (Fig. 2a), using the same antibodies that were also used in fixed cells (Emerin, BRG1 and H3K9me3). A clear advantage of the omission of crosslinking is the availability of certain downstream applications, which are not possible on cross-linked material. Furthermore, the protocol with non-fixed cells is less labor-intensive and less starting material is required. To avoid disruption of membrane structures during the procedure, we used the mild detergent digitonin for cell permeabilization. Digitonin is also used in recently developed epigenome profiling tools such as CUT & RUN[12]. Immunofluorescence experiments in non-fixed cells revealed expected biotinylation patterns using bait-specific antibodies, although not as clear as observed when using fixed cells (Fig. 2b). Subsequent streptavidin affinity enrichment of crude lysates revealed bait-specific interactomes of high quality, with many expected proteins and enriched GO terms (Fig. 2c, d; Supplementary Fig. 2a).

Having established efficient off-the-shelf proximity biotinylation strategies for the ProtA-Turbo enzyme in fixed and non-fixed cells, we aimed to further demonstrate the broad applicability of the ProtA-Turbo workflow. To this end, we performed H3K9me3-targeting experiments in a breast cancer cell line (MCF7 cells) and in myeloid leukemia cells (U937) (Supplementary Fig. 2b, c). These experiments, using fixed and non-fixed cells, respectively, revealed a range of known heterochromatin proteins, as well as many overlapping but also distinct H3K9me3 proximal proteins in both cell lines. We also performed ProtA-Turbo Emerin-targeting experiments in primary material, namely low passage primary human fibroblasts. This revealed correct biotinylation targeting of the nuclear envelope as assessed using immunofluorescence (Supplementary Fig. 2d). Subsequent mass spectrometry-based analyses revealed many known nuclear envelope components, thus demonstrating the applicability of the ProtA-turbo enzyme to

identify bait-proximal proteins in non-fixed, primary cells (Supplementary Fig. 2e–g). Finally, we evaluated the applicability of the ProtA-Turbo enzyme to study the effects of cellular perturbations. To this end, we performed H3K9me3 ProtA-Turbo-targeting experiments in wild-type mouse embryonic stem cells (ESCs) and ESCs lacking Suv39h1/2, the two enzymes responsible for depositing H3K9me3 on pericentromeric heterochromatin. As expected, immunofluorescence-based analysis revealed a reduction of H3K9me3 foci in Suv39h1/2 KO cells, indicative of a reduction of H3K9me3 on pericentromeric heterochromatin in Suv39h1/2 knock-out cells compared to wild type cells (Supplementary Fig. 2h). Importantly, Suv39h1/2 knock-out cells still contain significant levels of H3K9me3, catalyzed by Setdb1/2 and Ehmt1/2. ProtA-Turbo H3K9me3 targeting experiments in wild-type versus Suv39h1/2 knock-out cells confirmed a previously reported depletion of pericentromeric H3K9me3-proximal proteins, including Suv420h2, as well as other H3K9me3 reader proteins such as CDYL2 and UHRF1 in Suv39h1/2 knock-out cells compared to wild-type cells (Supplementary Fig. 2i), thus validating the applicability of our off-the-shelf proximity labeling workflow to study dynamic proximal proteomes upon cellular perturbations.

**FLYWCH1 localizes to centromeric, H3K9me3-marked chromatin.** ProtA-Turbo proximity biotinylation experiments revealed many known and previously unknown proximal proteins for the baits we investigated. As an example, in various ProtA-Turbo H3K9me3 targeting experiments we identified several unreported H3K9me3-proximal proteins and we confirmed some of these (RNF169, SENP1, and SENP7) by immunofluorescence and western blot (Supplementary Fig. 3a–c). In addition to these proteins, we identified FLYWCH1 as an H3K9me3-proximal protein. FLYWCH1 contains five so-called FLYWCH-type Zinc fingers, and has previously been identified as a regulator of beta-catenin signaling and is associated with various malignancies[13,14]. Furthermore, a homozygous Flywch1 deletion is embryonic lethal in mice[15]. However, it is currently unclear whether and how FLYWCH1 plays a role in chromatin-related processes. To study FLYWCH1 in more detail, we transiently expressed GFP-FLYWCH1 and we used CRISPR technology to generate endogenous GFP and V5 FLYWCH1 knock-in cell lines (Supplementary Fig. 3d, e). Immunofluorescence experiments revealed a strong overlap between FLYWCH1 and H3K9me3 in mammalian nuclei, consistent with ProtA-Turbo H3K9me3 proteomics experiments (Fig. 3a). Next, we performed ChIP-seq analysis for FLYWCH1 in duplicate using the cell line in which endogenous FLYWCH1 is tagged with GFP and a V5 tag (Supplementary Fig. 3f). These experiments revealed a significant genome-wide overlap between H3K9me3 and FLYWCH1 (Fig. 3b). DNA motif analyses amongst 451 high confidence FLYWCH1-binding sites in the genome (Supplementary Fig. 3g) revealed strong enrichment of simple repeats (Fig. 3c, d). In addition, many strong FLYWCH1-binding sites in

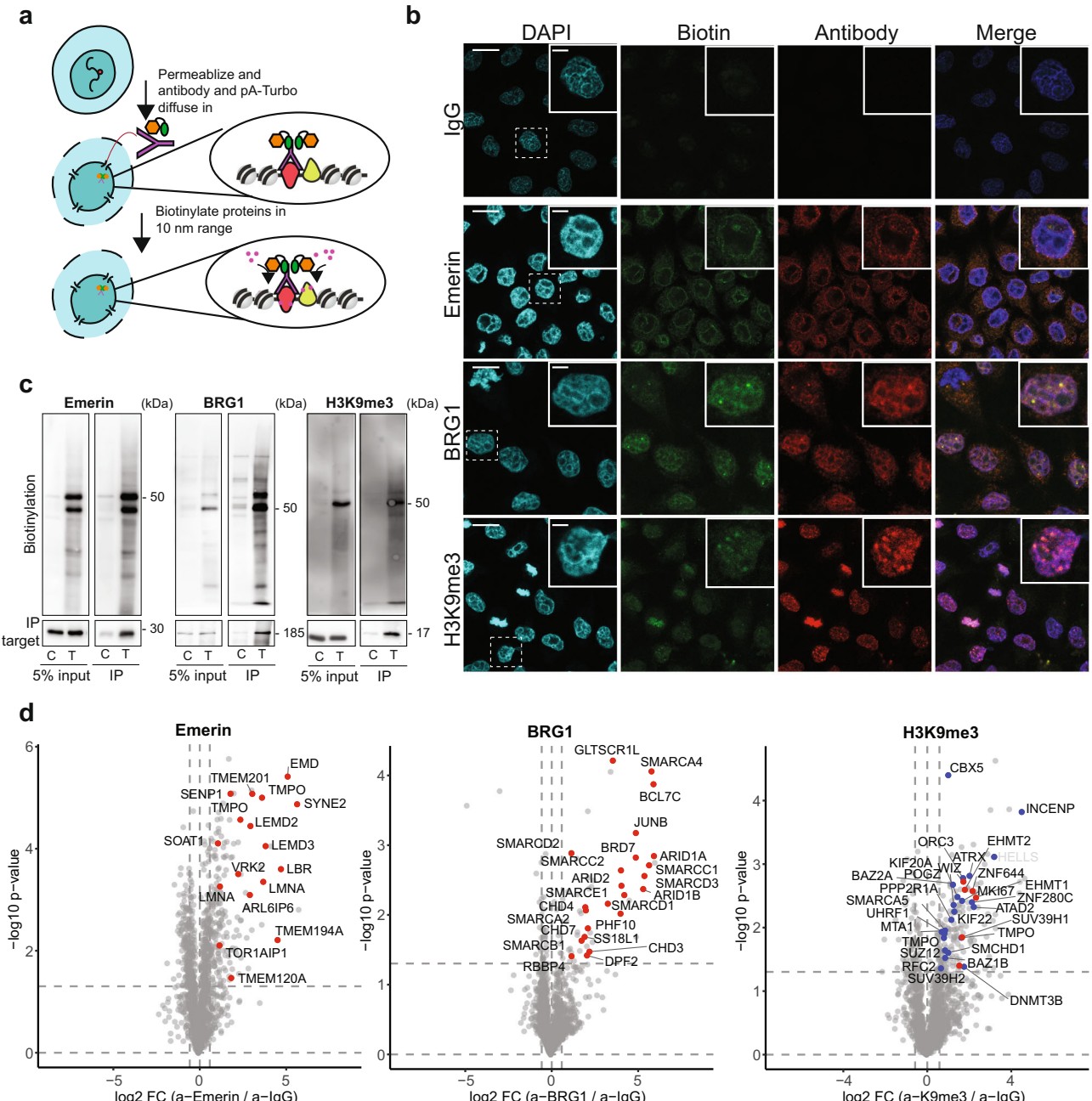

**Fig. 2 A ProteinA-TurboID fusion protein allows enrichment of protein localized to a specific sub-nuclear compartment in non-crosslinked cells.**
**a** Schematic outline of the method. **b** Immunofluorescence images of targeting the nuclear lamina (via Emerin), the SWI/SNF protein complex (via BRG1), or (pericentromeric) heterochromatin (via H3K9me3) in non-crosslinked HeLa cells. IgG was used as control antibody. Biotinylation (in green) overlapping the antibody signal (in red) illustrates the correct localization of the ProtA-Turbo fusion protein. Scale bars represent 10 μm (large panels) or 3 μm (small panels). **c** Immunoprecipitation of biotinylated proteins after targeting (T) proteins as in (**b**). IgG has used a control (C). **d** Volcano plot of mass spectrometry analyses of biotin IPs as in (**c**). A selection of proteins known to localize to the targeted proteins is highlighted. In the H3K9me3 volcano plot, red dots indicate writer/writer complexes, and blue indicates known pericentromeric proteins. Protein names in white indicate potential streptavidin contaminants (see also Source Data). the x-axis represents the log2 fold change (FC) and the y-axis the −log10 p-value of comparing three IgG replicates with three specific antibody replicates. Number of independent experiments performed: **b** 3 (IgG), 3 (Emerin), 2 (BRG1), 3 (H3K9me3); **c** 3 (Emerin), 2 (BRG1), 3 (H3K9me3); **d** 2 (Emerin), 1 (BRG1), 3 (H3K9me3), n = 3 replicates per independent experiment. Source data are provided as a Source Data file.

the genome are localized at centromeric heterochromatin, which is known to be enriched for H3K9me3 (Supplementary Fig. 3h). Given the observation that FLYWCH1 localizes to centromeric regions throughout the cell cycle in vivo (indicated by INCENP binding) (Supplementary Fig. 3e), and given the fact that these regions are strongly enriched for H3K9me3, we investigated

whether FLYWCH1 is a reader protein for the H3K9me3 modification. However, H3K9me3 peptide pulldowns in crude nuclear extracts indicated that FLYWCH1 does not interact with H3K9me3 directly (Supplementary Fig. 3i). Next, we performed DNA affinity purifications with several versions of putative FLYWCH1-binding motifs that we identified by ChIP-seq

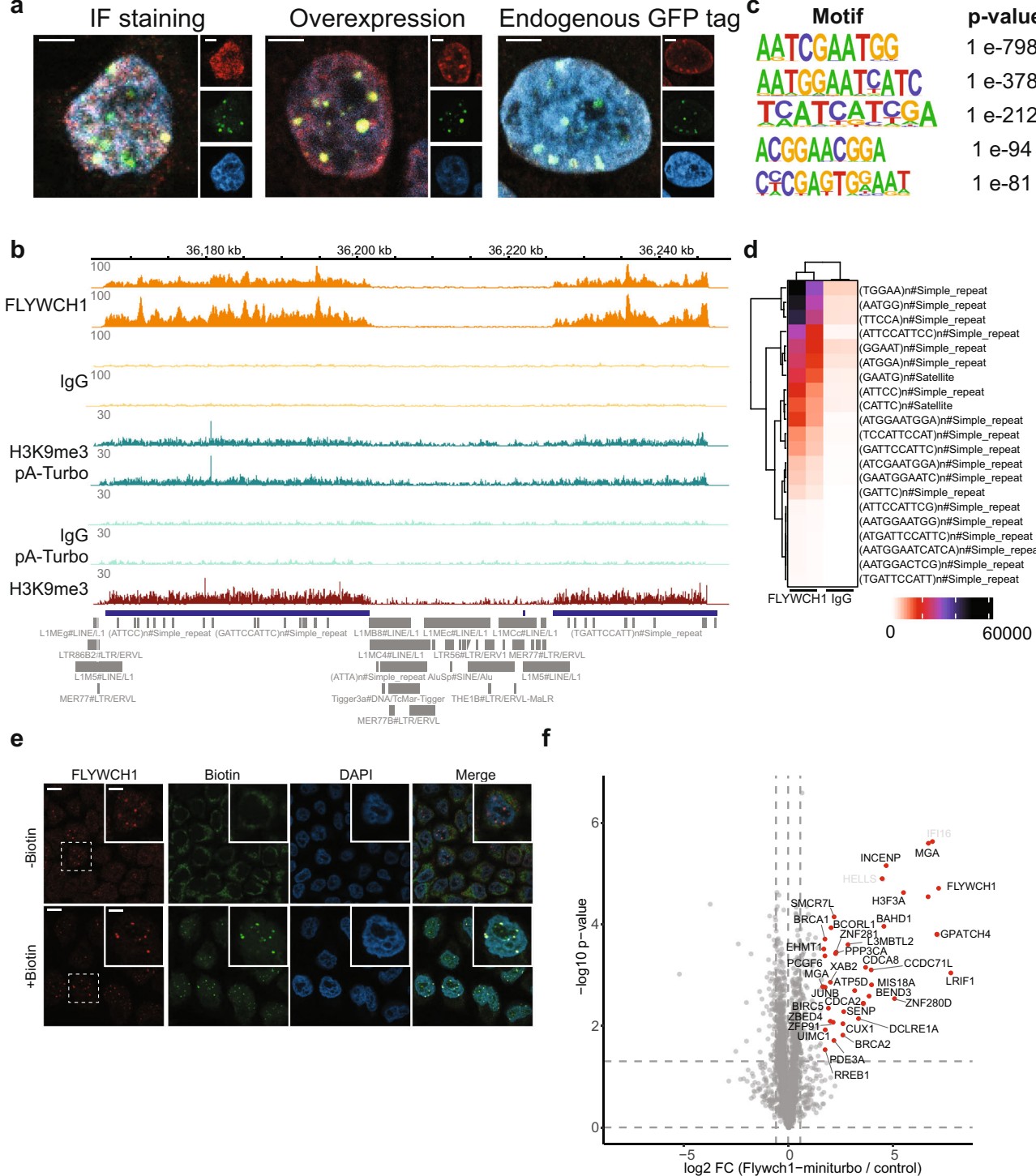

**Fig. 3 ProtA-Turbo targeting of H3K9me3 reveals FLYWCH1 as a protein localized to (peri)centromeric heterochromatin. a** Immunofluorescence of FLYWCH1 antibody staining (left), FLYWCH1-GFP overexpression (middle) or GFP-signal of FLYWCH1 endogenously tagged with GFP (right). The large panel is the merge, and the small panels are H3K9me3 (red), antibody/GFP signal (green), or DAPI (blue). Scale bars represent 3 μm. **b** Representative screenshot of FLYWCH1-bound genomic regions as assessed by ChIP-seq. IgG was used as a control antibody. Gray markers indicate different repeat types from the Repeatmasker. **c** Most enriched DNA motifs under 451 FLYWCH1-specific peaks. P-values were calculated using hypergeometric enrichment calculations using HOMER. **d** Intersection of FLYWCH1 ChIP-seq reads with most enriched repeat types/sequences that are at least 10-fold differential between FLYWCH1 and IgG ChIP-seq. **e** Immunofluorescence analyses of FLYWCH1 fused to miniTurboID. FLYWCH1 is visualized by staining the V5 tag (red). Scale bars represent 10 μm in the large panels and 5 μm in the zoomed panels. **f** Volcano plot of mass spectrometry analyses of biotin IPs of FLYWCH1-miniTurboID with biotin compared to wildtype HeLa cells incubated with biotin ($n = 3$). Protein names in white indicate potential streptavidin contaminants (see also Source Data). Number of independent experiments performed: **a** 2; **b** 2; **e** 2; **f** 2, $n = 3$ replicates per independent experiment. Source data are provided as a Source Data file.

(Fig. 3b), which revealed specific FLYWCH1 binding compared to scrambled versions of the motifs (Supplementary Fig. 3j). Selective binding of FLYWCH1 to certain DNA sequences at (peri)centromeric chromatin loci may explain the striking chromosome-specific-binding pattern we observed for FLYWCH1 using ChIP-seq and immunofluorescence.

To gain further insights into the function of the FLYWCH1 protein, we aimed to generate a knock-in cell line in which FLYWCH1 is directly fused to a proximity biotinylation enzyme. While efforts to generate a FLYWCH1-TurboID knock-in cell line were unsuccessful, we obtained a cell line in which endogenous FLYWCH1 is fused to miniTurboID, which is a smaller and less active version of the TurboID enzyme.[7] Immunofluorescence experiments revealed that nuclear biotinylation in this FLYWCH1-miniTurbo cell line is localized in punctuated foci, which is consistent with the H3K9me3-staining pattern in mammalian cells (Fig. 3e). Subsequent streptavidin-based affinity enrichment experiments from miniTurbo-FLYWCH1 cells revealed numerous FLYWCH1 proximal proteins (Fig. 3f and Supplementary Fig. 3k). These FLYWCH1-proximal proteins include the H3K9me3 methyltransferase EHMT1, the HP1 interactor MGA, subunits of the chromosomal passenger complex (INCENP, CDCA8), and Polycomb proteins. In summary, these experiments uncovered FLYWCH1 as an H3K9me3 proximal protein that localizes to a subset of (peri)centromeres in human cells, together with many additional H3K9me3-associated proteins. Future experiments using FLYWCH1 perturbations experiments are needed to further uncover a putative role for FLYWCH1 in centromere maintenance or regulation and cellular homeostasis.

## Discussion

In this study, we present an enzyme, called ProtA-Turbo, which can be used for proximity biotinylation and interaction proteomics purposes without the requirement for genetic manipulation or transfection of target cells. We illustrate the applicability of the ProtA-Turbo enzyme by targeting three nuclear baits using polyclonal antibodies and we show that the enzyme can be used for interaction proteomics studies using fixed and non-fixed cells. Finally, we illustrate the usefulness of the ProtA-Turbo enzyme through the identification and initial characterization of a relatively uncharacterized protein, FLYWCH1, as a marker of centromeric, H3K9me3-marked chromatin.

Over the years, a range of methods has been developed to determine the proximal proteome of a protein of interest. ChIP-MS-based approaches (i.e. antibody-based enrichment of chromatin fragments) are solely applicable to chromatin-bound proteins, require sonication, which is a highly variable process[16], and take several days to perform, thus limiting throughput[8]. Proximity biotinylation requires transfection or genetic manipulation, which potentially induces biological changes in the cell type of interest. Furthermore, genetic manipulation of target cells is labor-intensive and is not applicable to all types of cells, such as non-proliferative primary cells. As an alternative, antibody-HRP (horseradish peroxidase) conjugates can be used to biotinylate local environments, but these methods are restricted to fixed material since HRP is denatured in the acidic environment of live cells[17,18]. The fixed version of the off-the-shelf method presented here extends on the principle of the antibody-HRP conjugates, achieves higher enrichment of target-site specific GO terms, and is less labor-intensive (cf. BAR, Fig. S1j). Both HRP and ProtA-Turbo-based methods rely on the specificity of the used antibodies as well as the abundance of the proteins or modifications that are targeted. ProtA-Turbo-targeting experiments on low abundant proteins or modifications may suffer from high background biotinylation signal since at any given time a number of

ProtA-Turbo molecules may not be target engaged. 'Classical' proximity biotinylation experiments in which the bait of interest is directly fused to a biotinylation enzyme do not suffer from these drawbacks. On the other hand, fusing a proximity biotinylation enzyme to a bait protein of interest may negatively affect the function of that bait. Furthermore, PTMs are difficult to target using classical proximity biotinylation experiments but these can easily be targeted using the ProtA-Turbo enzyme. Thus, classical proximity biotinylation experiments and ProtA-Turbo mediated proximity biotinylation experiments each have their own pros and cons and are complementary in nature.

While off-the-shelf proximity biotinylation on cross-linked cells enriches the local proteome of a protein of interest, the usage of formaldehyde-based crosslinking precludes certain downstream applications. A notable example comprises crosslinking-mass spec (XL-MS), which can be used to discriminate directly from indirect protein–protein interactions[19]. This approach almost exclusively targets unmodified lysine residues[20], but formaldehyde fixation of cells also targets these lysines. To allow future studies aimed at combining XL-MS with proximity biotinylation, omission of crosslinking would be beneficial. The native off-the-shelf workflow developed in the current study thus allows additional downstream workflows, including assessing protein–protein interaction topologies using XL-MS. In addition, we noticed that some cell types, such as U937 cells, tend to clump after crosslinking, resulting in loss of material. The native workflow circumvented these issues and allowed to obtain of a comprehensive H3K9me3 proximal proteome in U937 cells. Another advantage of the native version is that it is less labor-intensive compared to the fixed protocol. The crosslinking-based and native workflows yield comparable results. However, we do observe some technical variability between experiments concerning, for example, fold enrichments for bait-proximal proteins. We believe these variations are technical in nature related to for example small differences in washing conditions and generated lysates. Use of the cross-linking-based versus the native method will depend on potential downstream applications and target cells of interest. Both approaches enrich a range of expected proteins from targeted subcellular compartments. However, it is conceivable that fixation and/or permeabilization may have mild effects on protein complex formation or localization. Therefore, orthogonal methods (such as western blotting, immunofluorescence, or ChIP-seq) should be applied to validate newly identified proximal proteins for baits of interest. Finally, affinity purifications from crude lysates using commonly used affinity enrichment reagents such as Flag, HA, and streptavidin beads suffer from false-positive contaminants, which are reported in the so-called CRAPome repository[21]. The streptavidin-based affinity enrichments we performed in this study also yielded a number of proteins that are listed in the CRAPome repository as potential false positive hits (also see Source Data). These issues might be partially overcome by mapping biotinylated peptides in bait proximal proteins, as reported previously[22–24]. However, in the experimental workflow used here with partial tryptic digestion on beads followed by further digestion in the solution overnight, biotinylated peptides remain bound to streptavidin beads (streptavidin bound to the beads does not get completely digested during a brief trypsin incubation at RT), while other tryptic peptides are released from the beads, preventing identification of biotinylated peptides in the mass spectrometer.

In the current study, we focused on targeting the ProtA-Turbo enzyme to nuclear proteins and a transcriptionally repressive histone modification. Future applications of the ProtA-Turbo enzyme will include targeting cytoplasmic and cell-surface proteins, for example in the context of cancer immunotherapy. Furthermore, modifications on nucleic acids such as DNA or

RNA methylation can be targeted using commercially available high-quality antibodies against these modifications. Other options include fusing TurboID to specific chromatin reader domains to generate an off-the-shelf alternative to the recently developed ChromID technology[11]. Together, these off-the-shelf approaches provide a highly flexible toolbox to perform proximity biotinylation assays in any cell type of interest in a fast and efficient manner.

## Methods

**Cell culture.** HeLa (source: ATCC) and MCF7 (source: ATCC) cells were cultured in Dulbecco's modified Eagle medium (DMEM, Gibco) supplemented with 10% FBS and penicillin/streptomycin. U397 cells (source: ATCC) were cultured in RPMI with 10% FBS and penicillin/streptomycin. Low passage primary fibroblasts from Coriell institute (AG08469) were cultured with DMEM supplemented with 15% FBS and penicillin/streptomycin and used until passage 12. Mouse ESCs Suv39h1/2 WT and double KO were a kind gift from Dr. Thomas Jenuwein and were cultured in DMEM supplemented with 15% FBS, penicillin/streptomycin, 10 mM sodium pyruvate, 5 μM beta-mercaptoethanol, and leukemia inhibitory factor (1000 U/ml).

**Primers and oligonucleotides.** A full list of primers and oligonucleotides used in this work is found in Supplementary Table 1.

**Recombinant protein purification.** The MNase sequence in the pk19pAMNase vector (pk19pAMNase was a gift from Ulrich Laemmli[25], Addgene #86973) was substituted by the TurboID sequence. TurboID was amplified from the plasmid 3xHA TurboID NLS pcDNA3 (3xHA TurboID NLS pcDNA3 was a gift from Alice Ting[7], Addgene #107171) and inserted in the pK19 vector between EcoRI and BamHI restriction enzymes using the primers TurboEcoRI-F, TurboBamHIfrag-R, TurboBamHIfrag-F, and TurboBamHI-R. A histidine stretch was added later to the N-terminal part of the protein using the oligos 6xHistagHinDIII-F and 6xHistagHinDIII-R to facilitate purification of the enzyme.

The Pk19 ProtA-Turbo plasmid was transformed into C3013 E. coli. Colonies were grown overnight and cultured in 2 l LB medium until reaching OD 0.6. Protein expression was then induced for 3 h with 2 mM IPTG. The bacteria were harvested, resuspended in 70 ml of lysis buffer (0.5 M NaCl, 10% glycerol, 20 mM HEPES pH = 8, 1 mM EDTA pH = 8, 0.1% NP40, 20 mM b-mercaptoethanol, and 1 mM PMSF) and sonicated. The lysate was centrifuged for one hour at $12,000 \times g$ at 4 °C to eliminate debris. The protein extract was then allowed to pass through a column (Econo-Pack Disposable Chromatography Columns, 10 ml, BioRad) containing 2 ml Ni-NTA agarose beads (Qiagen) by gravity flow, and the column was then washed once with lysis buffer and twice with wash buffer (10 mM Tris, 0.5 M EDTA, 10% Glycerol, 500 mM NaCl). A pre-elution was performed with 5 ml of wash buffer containing 15 mM Imidazol and 1 ml fractions were collected. The majority of the ProtA-Turbo enzyme was then eluted with 10 ml of wash buffer containing 100 mM Imidazol, also collected in 1 ml fractions. Aliquots of the collected fractions were loaded on an SDS–PAGE gel and stained with Imperial Protein Staining (Thermo Scientific) following the manufacturer's instructions to determine the amount and purity of the enzyme. Fractions were pooled according to their similarity on gel and were snap-frozen and stored at −80 °C until further usage.

**ProtA-Turbo immunofluorescence.** Cells were cultured in coverslips (15 mm diameter) in a 12-well plate. The next day, cells were washed with PBS and, after that, fixed in PBS containing 4% v/v PFA for 15 min at RT, washed three times with PBS, permeabilized in PBS containing 0.3% v/v Triton for 10 min, and blocked in blocking solution (3% BSA in 0.3% Triton-PBS) for 30 min. Coverslips were then incubated for one hour in a humid chamber with primary antibody diluted 1:150 in blocking solution (H3K9me3 antibody (Abcam, ab8898), BRG1 antibody (A300-813A, Bethyl), and Emerin antibody (10351-1-AP, Proteintech)). Coverslips were washed four times with PBS and incubated with 140 ng of ProtA-Turbo enzyme diluted in 30 μl of blocking buffer per coverslip for one hour and then washed 4 times with PBS. Coverslips were then incubated with biotin reaction buffer (5 mM MgCl₂, 5 μM Biotin, 1 mM ATP in PBS) for 10 min at 37 °C, followed by three washes with PBS. Finally, coverslips were incubated with secondary antibody (anti-rabbit Alexa fluor 568 (Life Technologies A11004, dilution 1:1000), FITC-Avidin (ThermoFisher Scientific A821, 1:200) and DAPI for one hour, washed 4 times with PBS and fluoromount G (Thermo, 00-4958-02) was used for mounting the slides. Images were acquired with a confocal microscope LSM900 (ZEISS) and analyzed with Fiji software.

**Prot A-Turbo targeting in fixed cells.** Cells were cultured in 15 cm dishes. Next day, cells were washed once with PBS and then fixed in 3 ml of PBS containing 4% v/v PFA for 15 min at RT. Cells were then collected using a cell scraper. After fixation cells were centrifuged at $150 \times g$ for 3 min at RT. At least 70 μl of cell pellet was used for each antibody. Cell pellet was resuspended in 1 ml of hypotonic lysis

buffer (10 mM Tris pH 7.5, 10 mM NaCl, 3 mM MgCl₂, 0.3% NP40, and 10% glycerol), incubated for 10 min on ice, and then centrifuged at $800 \times g$ for 8 min at 4 °C to isolate the nuclei. Nuclei were then washed three times with 0.5 ml of hypotonic lysis buffer and centrifuged at $200 \times g$ for 2 min at 4 °C. Nuclei were then washed once with 0.5 ml of PBS. From this step, centrifugations were performed at RT between 100 and $900 \times g$ (required centrifugation speed to pellet cells varies depending on the cell type) for 3 min. Nuclei were resuspended and permeabilized in 1 ml of PBS supplemented with 0.3% v/v Triton-X100 and incubated in a rotation wheel for 10 min. Non-specific-binding sites were blocked with 1 ml PBS containing 0.3% Triton-X100 supplemented with 3% BSA for 30 min in a rotation wheel at RT. Samples were then incubated with 3 μg of primary antibody diluted in 300 μl of blocking solution (IgG rabbit antibody (12-370, EMD Millipore), H3K9me3 antibody (Abcam, ab8898), BRG1 antibody (A300-813A, Bethyl), and Emerin antibody (10351-1-AP, Proteintech)) in a rotation wheel for one hour at RT. To remove the unbound antibody, nuclei were washed twice with 500 μl of PBS. Next, nuclei were incubated with 3.4 μg of ProtA-Turbo enzyme diluted in 300 μl blocking solution for one hour in a rotation wheel at RT. The unbound ProtA-Turbo was then removed by washing the pellet twice with 500 μl of PBS. Next, samples were incubated with 300 μl of biotin reaction buffer (5 mM MgCl₂, 5 μM Biotin, 1 mM ATP in PBS) in a thermo shaker at 1000 rpm for 10 min at 37 °C. Samples were then centrifuged, washed once with 1 ml PBS and finally lysed in 300 μl of RIPA buffer (50 Mm Tris pH 7.8, 150 mM NaCl, 0.5% sodium deoxycholate, 0.1% SDS, 1% NP40). overnight on ice.

The following day, samples were sonicated using a Bioruptor sonicator (Diagenode) (6–10 cycles of 30 s ON 30 s off, maximum power) until the samples became more transparent and then decrosslinked for one hour at 95 °C after addition of SDS to a final concentration of 1%. After an additional sonication cycle, samples were centrifuged at $21,000 \times g$ for 10 min at 4 °C. Supernatant was then recovered and incubated with 12.5 μl streptavidin sepharose high-performance beads (15511301, Cytiva) on a rotation wheel for 2 h at 4 °C. Beads were then washed 5 times with 0.5 ml RIPA buffer. At this point in the protocol, if the purpose was to perform a Western Blot, 40 μl of protein loading buffer (125 mM Tris pH 6.8, 25% glycerol, 5% SDS, 0.1% bromophenol blue, 1.43 M b-mercatoethanol) was added to the beads after which they were boiled at 95 °C for 10 min. Samples were then loaded on an SDS–PAGE gel and further processed for western blotting. If the purpose was to analyze samples by mass spectrometry, the experiments were performed in triplicate for each antibody and, after 5 washes with 500 μl of RIPA buffer, the beads were washed 4 times with 500 μl of PBS. 50 μl of elution buffer (2 M Urea, 10 mM DTT, 100 mM Tris pH 8) was then added to the beads, followed by incubation on a thermo shaker at 1500 rpm for 20 min at RT. Iodoacetamide was then added to the samples to a final concentration of 50 mM and samples were incubated at 1500 rpm in the dark for 10 min at RT. 2.5 μl of trypsin (0.1 mg/ml trypsin stock solution) was then added to each sample, followed by incubation in a thermo shaker at 1500 rpm for 2 h at RT. Samples were then centrifuged and the eluates were collected. 50 μl of elution buffer was then added to the beads and after incubation in a thermo shaker at 1500 rpm for 5 min at RT, the eluates were collected and combined with the first elution fraction. 1 μl of fresh trypsin was then added to each sample and samples were incubated overnight at RT. The following day, peptides were acidified by adding 10 μl of 10% v/v TFA and purified on C18 Stagetips (3M Empore)[26]. Peptides on Stagetips were stored at 4 °C until being subjected to LC–MS.

When H3K9me3 (Abcam, ab8898) or Emerin antibody (10351-1-AP, Proteintech) was used, isolation of nuclei was performed before fixation. Briefly, cells were washed with PBS, collected and transferred to an Eppendorf. Cells were centrifuged at $400 \times g$ for 3 min at RT after which the supernatant was removed. Cell pellet was then resuspended in 1 ml of hypotonic lysis buffer (10 mM Tris pH 7.5, 10 mM NaCl, 3 mM MgCl₂, 0.3% NP40, and 10% glycerol), incubated on ice for 10 min, and then centrifuged at $800 \times g$ for 8 min at 4 °C. Nuclei were then washed three more times with 500 μl of lysis buffer and once with PBS followed by centrifugation at $200 \times g$ for 2 min at 4 °C. Nuclei were then fixed in 1 ml of PBS containing 4% v/v PFA and incubated in a rotation wheel for 15 min at RT. Finally, nuclei were washed twice with PBS and permeabilized with 1 ml of PBS containing 0.3% v/v Triton-X100 and incubated for 10 min in a rotation wheel at RT. From this step, samples were processed as described for the other antibodies.

**ProtA-Turbo targeting in unfixed cells.** Cells were washed with PBS and collected from the plates using a cell scraper. Then, cells were centrifuged at 150 g for 3 min at RT. Around 70 μl of cell pellet was used for each replicate. Cell pellets were incubated with 500 μl of digitonin buffer (0.04% digitonin diluted in 20 mM HEPES KOH pH 7.5, 150 mM NaCl, 0.5 mM Spermidine) in a rotation wheel for 10 min at RT. A small aliquot of sample was then stained with trypan blue and analyzed using a Countess Cell Counter (Invitrogen) for cell viability. Samples were further processed when cell viability was <20%, thus ensuring sufficient outer membrane permeabilization. If cell viability was >20%, more digitonin buffer or a higher digitonin concentration was added to obtain >80% non-viable cells. Next, cells were centrifuged at $100–900 \times g$ (required centrifugation speed to pellet cells varies depending on the cell type) for 3 min at RT. 2 μg of antibody (IgG, Emerin, BRG1, and H3K9me3) diluted in 200 μl digitonin buffer was then added to the samples. Samples with antibodies were incubated for 20 min in a thermo shaker at 600 rpm at RT. Cells were then washed twice with digitonin buffer and incubated

with 1.4 μg of ProtA-Turbo enzyme diluted in 500 μl digitonin buffer in a rotation wheel for 30 min at 4 °C. Cells were then washed twice with digitonin buffer. Next, cells were incubated with biotin reaction buffer (5 mM $MgCl_2$, 5 μM Biotin, 1 mM ATP in digitonin buffer) in a thermo shaker at 1000 rpm for 10 min at 37 °C. Cells were washed once with 0.5 ml wash buffer without digitonin, resuspended in 300 μl RIPA buffer (50 Mm Tris pH 7.8, 150 mM NaCl, 0.5% sodium deoxycholate, 0.1% SDS, 1% NP40), and incubated overnight on ice. The following day, the samples were sonicated in an NGS Bioruptor (Diagenode) until the samples appeared transparent. Samples were then centrifuged at $21,000 \times g$ for 10 min at 4 °C after which the supernatant was incubated with 12.5 μl Streptavidin Sepharose High-Performance beads (15511301, Cytiva). From this step onwards, the procedure is the same as described for the protocol with fixed cells.

When the H3K9me3 or Emerin antibodies were used in HeLa cells, nuclei were isolated before antibody incubation. Briefly, cells were washed with PBS, collected using a cell scraper, and transferred to an Eppendorf. After centrifugation at $100 \times g$ for 3 min, the cell pellet was resuspended in 1 ml of hypotonic lysis buffer (10 mM Tris pH 7.5, 10 mM NaCl, 3 mM $MgCl_2$, 0.3% NP40, and 10% glycerol), incubated for 10 min on ice and then centrifuged at $800 \times g$ for 8 min at 4 °C. Nuclei were then washed three times with the same buffer and centrifuged at $200 \times g$ for 2 min at 4 °C. Finally, cells were washed once with wash buffer (no digitonin) and incubated with 2 μg antibody diluted 200 μl digitonin buffer. From this step samples were processed as described for the other antibodies.

**LC-MS/MS measurements.** Digested peptides were eluted from C18 Stagetips with 30 μl of buffer B (0.1% formic acid, 80% acetonitrile) and after speedvac centrifugation to 5 μl, buffer A (0.1% formic acid) was added to a total volume of 12 μl. 6 μl of each sample was then loaded on an Easy-nLC1000 (Thermo) connected online either an LTQ-Orbitrap-Fusion (Thermo), an LTQ-Orbitrap Q-Exactive HFX mass spectrometer (Thermo) or an Orbitrap Exploris (Thermo). The LC–MS and data acquisition method used for the LTQ-Orbitrap-Fusion were: a gradient from 9% to 32% buffer B for 114 min before washes at 50% then 95% buffer B were performed, for 140 min of total data-collection time. Scans were collected in data-dependent top speed mode with dynamic exclusion set at 60 s. For the LTQ-Orbitrap Q-Exactive, peptides were separated with a 94-min gradient from 9% to 32% buffer B followed by washes at 50% and then 95% buffer B for 120 min of total data-collection time. All scans were collected in data-dependent mode and collected in a top10 data-dependent acquisition mode[27–29]. For the Orbitrap Exploris, an acetonitrile gradient of 12–30% in 43 min was used, after which the acetonitrile concentration was increased to 60% in 10 min and then up to 95% in 1 min. Total data acquisition time was 60 min. The spray voltage was set to 2200 V in positive ion mode. The expected LC peak width was 15 s. The full scan of the peptides was set to a resolution of 120,000 in a scan range of 350–1300 $m/z$. The normalized AGC target was 300% and the maximum injection time was set to 20 ms.

**LC-MS/MS data analyses.** Raw files were analyzed using standard settings of MaxQuant 1.5.1.0. Options LFQ, iBAQ, and match between runs were enabled. As a search database, the human fasta database updated in 2017 from Uniprot was used. Perseus 1.5.0.15 was used to filter proteins flagged as a contaminant, reverse or only identified by site. All of the experiments were performed in triplicate for each antibody, using an independent cell mixture for each replicate as input. The triplicates were grouped based on experimental conditions (based on the primary antibody used) and only proteins that had an LFQ value in each of the replicates in at least one group of triplicates were maintained for downstream analyses. Next, missing values were imputed using default parameters in Perseus. Statistically different proteins were identified using a Student's $t$-test (FDR < 0.05) between targeted bait triplicates and IgG control triplicates. Additionally, a protein was called enriched only when it also had at least a 1.5-fold change over the IgG controls. The volcano plots were generated by plotting the $-\log10$ of the $p$-value, calculated by a Student's $t$-test comparing the bait-triplicates with IgG triplicates, against the fold change (which corresponds to the average fold change between bait-triplicates and IgG triplicates). Downstream data visualization was performed in R. Proteins that are known to localize to H3K9me3 domains, nuclear lamina, or SWI/SNF complexes were determined using literature curation. Pericentric heterochromatin binding proteins were obtained from pericentromeric purifications in mouse ESCs[30]. GO-analyses were performed using Clusterprofiler v4.0.0[31]. For GO-based comparison of ProtA-Turbo Emerin with other published lamin-targeting strategies, we used data for nuclear lamina microdomain mapping (LAP2B-BioID[32]), a two-component lamina BioID mapping strategy (2C-BioID[33]), conventional Lamin A BioID[3], and an antibody-HRP-based method for biotinylation by antibody recognition (BAR[17]). Enriched proteins were defined according to the following criteria: LAP2B-BioID > 1.5-FC/ctrl; 2C-BioID > 1.5-FC/ctrl (-noAP21967); Lamin A BioID: proteins were downloaded from Table S1; BAR: all proteins of the LMNA Unbound sample that are >2 fold enriched over control, were not shared with the no antibody control and had at least 2 unique peptides. All GO terms were determined using Clusterprofiler and 10 GO terms centered around the nuclear lamina were selected for evaluation of method performance. For the interaction network, proteins were selected that were significantly enriched in either H3K9me3 enrichments (HeLa fixed, HeLa unfixed, MCF7 fixed, and U937

unfixed) and showed at least a two-fold enrichment over IgG control. Proteins present in at least three of these datasets were retained and queried using the stringapp plugin of Cytoscape 3.8.2[34]. Disconnected nodes, and nodes that had no interaction as determined using experimental evidence or documented in a database, were removed. All proteins that were enriched in either of the fixed or unfixed ProtA-Turbo experiments were searched in the Crapome database[21] against all proximity biotinylation experiments performed on human cells. Proteins that are both enriched in ProtA-Turbo experiments with different baits relative to IgG (indicating potential non-specific enrichment), and that are frequently found in the Crapome database with a relatively high average spectral count, are more likely to be contaminant proteins than those with specific enrichment for one bait and a low frequency and low average spectral counts in the Crapome database (see also Source Data).

**ChIP-sequencing.** Cells were crosslinked with 1% formaldehyde for 10 min at RT and then quenched with 125 mM glycine for at least 5 min at RT. Cells were washed with PBS, collected using a cell scraper, and pelleted. Cell pellets were resuspended in 5 ml buffer B (0.25% TritonX-100, 10 mM EDTA, 0.5 mM EGTA, and 20 mM HEPES) per 15 cm dish, collected, and centrifuged at $1000 \times g$ for 5 min at 4 °C. The resulting pellet was incubated with 30 ml of buffer C (150 mM NaCl, 1 mM EDTA, 0.5 mM EGTA, and 50 mM HEPES) in a rotation wheel for 10 min at 4 °C and centrifuged again at $1000 \times g$ for 5 min at 4 °C. The resulting pellet consisted of isolated nuclei. The nuclei were resuspended in 1 ml of incubation buffer (0.15% SDS, 1% Triton, 150 mM NaCl, 1 mM EDTA, 0.5 mM EGTA and 20 mM HEPES) and sonicated in a Bioruptor Pico sonicator (Diagenode). Chromatin fragment size was checked by decrosslinking 5 μl of sample and running this on an agarose gel. Most chromatin fragments were around 300 bp. 300 μl of chromatin was incubated with 3 μl of V5 antibody (P/N 46-0705, Invitrogen) in a rotation wheel overnight at 4 °C, together with a mix of 7.5 μl of protein A Dynabeads (10008D, Invitrogen) and 7.5 μl protein G Dinabeads (10009D, Invitrogen). Beads were then washed twice with Buffer 1 (0.1% SDS, 0.1% NaDOC, 1% TritonX-100, 150 mM NaCl, 1 mM EDTA, 0.5 mM EGTA and 20 mM HEPES), once with Buffer 2 (0.1% SDS, 0.1% Na doxycholate, 1% TritonX-100, 500 mM NaCl, 1 mM EDTA, 0.5 mM EGTA and 20 mM HEPES), once with buffer 3 (0.5% NaDOC, 0.5% NP40, 250 mM LiCl, 1 mM EDTA, 0.5 mM EGTA and 20 mM HEPES) and twice with buffer 4 (1 mM EDTA, 0.5 mM EGTA, 20 mM HEPES). For every wash, the beads were incubated with washing buffer in a rotation wheel for 5 min at 4 °C. After the washes, chromatin was eluted by adding 200 μl of elution buffer to the beads (1% SDS and 0.5 mM $NaHCO_3$) followed by incubation in a rotation wheel for 20 min at RT. The supernatant containing the eluted chromatin was decrosslinked by adding 8 μl of 5 M NaCl and 2 μl of 10 mg/ml Proteinase K and incubation in a thermo shaker at 1000 rpm for at least 4 h at 65 °C. DNA was purified using MiniElute columns (Qiagen). The library for sequencing was prepared with a Kapa HyperPrep Kit (Kapa Biosystems) following the manufacturer's instructions. NEXTflex adapters (Bio Scientific) were used during the sample prep. Samples were analyzed on an Agilent 2100 Bioanalyzer for purity and sequenced on an Illumina NextSeq500.

**ProtA-Turbo ChIP-sequencing for H3K9me3.** Cells were first subjected to the ProtA-Turbo protocol described for fixed cells using the H3K9me3 antibody. After washing away the biotinylation reaction buffer, nuclei were resuspended in 1 ml of SDS buffer (50 mM Tris pH 8, 0.5% SDS, 100 mM NaCl, 5 mM EDTA) and incubated for 10 min on ice. Nuclei were then pelleted and resuspended in 300 μl of IP buffer (0.3% SDS, 1.1% Triton, 1.2 mM EDTA, 16.7 mM Tris pH 8, 167 mM NaCl) and sonicated in a Bioruptor Pico Sonicator. The samples were precleared with 30 μl Dynabeads protein A and 5 μl of 5% v/v BSA (final concentration ~0.05%) in a rotation wheel for 1 h at RT. Next, samples were incubated with 30 μl of Streptavidin M280 beads (Invitrogen) for 3 h at RT. Beads were then washed twice with 2% SDS and three times with LiCl buffer (100 mM Tris pH 8, 500 mM LiCl, 1% NP40, 1% sodium deoxycholate). Samples were then incubated in 60 μl of 300 mM NaCl in a thermo shaker at 1000 rpm overnight at 65 °C. Following day, 15 μl of proteinase K buffer (50 mM Tris pH 7.5, 25 mM EDTA, 1.25% SDS) and 1.5 μl proteinase K (10 mg/ml) was added to the sample followed by incubation in a thermo shaker for 2 h at 45 °C to ensure complete decrosslinking. The supernatant was then collected. DNA was purified using MiniElute columns (Qiagen). Library preparation for sequencing was performed as described for FLYWCH1 ChIP-sequencing.

**ChIP-sequencing analysis.** ChIP-seq libraries were sequenced paired-end on an Illumina Nextseq 500 sequencer. For biotin ChIP-seq after ProtA-Turbo targeting using H3K9me3 antibody or IgG control, the reads were mapped against the hg38 genome build using bwa 0.7.17 with parameters mem -t 32[35]. Reference H3K9me3 ChIP-seq in wild-type HeLa cells was downloaded from GEO (accession number GSE86814[36]) and processed in parallel. Duplicate reads were identified and filtered using Picard tools version 1.129 (http://broadinstitute.github.io/picard/). FLYWCH1 and IgG ChIP-seq were processed using the seq2science pipeline[37]. Parameters were fastp as trimmer, bwa-mem2 as aligners, minimal map quality of 30. For all files, peaks were called using macs2 2.2.7.1[38] with $q$-value 0.001 for

H3K9me3 tracks and *q* 0.05 for FLYWCH1 and IgG ChIP-seq. Bigwig files were visualized in the Integrative Genomics Viewer[39]. ChIP-seq heatmaps were generated using fluff 3.0.3[40]. Motif analysis was performed using Homer v4.11[41] using default parameters, genome hg38, and width 200. Karyotype plots were generated in R using karyoploteR 1.18.0[42]. For intersection with repeat elements, the repeat masker database was downloaded using the UCSC table browser. Reads intersecting the repeat coordinates were obtained using tools 2.27.1 multicov. Only repeat regions that had more than 1000 reads and were at least 10-fold different between FLYWCH1 and IgG ChIP-seq were retained.

**Immunofluorescence**. Immunofluorescence was performed using the same protocol as described for the ProtA-Turbo protocol with the omission of protA-Turbo addition and biotinylation. The following primary antibodies were used: α-H3K9me3 (Abcam, ab8898 and Active Motif, 61014), α-FLYWCH1 (Novus Biologicals, NBP1-85041), α-FLAG (Sigma, F3165), and α-INCENP (Active Motif, 39260) all in a 1:150 dilution.

**FLYWCH1, RNF169, SENP1, and SENP7 overexpression**. EGFP FLYWCH1 was cloned in the pEGFP-C3 vector. FLYWCH1 cDNA was amplified using as a template the vector IRATp970F08101D from Source Bioscience with the primers FLYWCH1EcoRI-Fw and FLYWCH1KpnI-Rv. Both vector and PCR products were cut by EcoRI and KpnI and ligate. FLAG-RNF169, FLAG SENP1, and FLAG-SENP7 was overexpressed using the plasmid pcDNA5-FRT/TO-Flag-RNF169 (this plasmid was a gift from Daniel Durocher[43], Addgene plasmid # 74243), the plasmid Flag-SENP1 (this plasmid was a gift from Edward Yeh[44], Addgene plasmid # 17357), and plasmid Flag-SENP7 (a gift from Edward Yeh[45], Addgene plasmid # 42886), respectively. HeLa cells were transfected with PEI and harvested 48 h after transfection or fixed for immunofluorescence analyses.

**Knock-in cell lines**. The pUC57 modified plasmid (a kind gift from Jop Kind lab) and the pU6-(BbsI)-Cbh-Cas9-T2A-mCherry (pU6-(BbsI)-Cbh-Cas9-T2A-mCherry plasmid (a gift from Ralf Kuehn[46], Addgene #64324) were used for tagging endogenous FLYWCH1 and Emerin. For FLYWCH1, the gRNA (GGGTGCTGAGCGTG GCCTGA) was inserted in the pU6-(BbsI)-Cbh-Cas9-T2A-mCherry and it targets the 5′ end of the FLYWCH1 sequence. FLYWCH1 homology arms were inserted in both sites of BSD-P2A-GFP-V5 or BSD-P2A-MiniturboID-V5. The gBlock HA1FLYW CH1 was inserted as a homology arm in the vector and the primers HA2FLYWCH1 NotI-Fw and HA2FLYWCH1NotI-Rv were used for the amplification of one of the homology arms from cDNA. The primers TurboNheIfrag-F, TurboNheIfrag-R, TurboNotI-R, and MiniTurbogoodNheI-F were used to amplify MiniturboID from the vector 3xHA MiniTurboID NLS pcDNA3 (3xHA miniTurboID NLS pcDNA3 was a gift from Alice Ting[7], Addgene #107172) and replace GFP and the gBlock 3xV5 to add V5 and a flexible linker as a tag. For Emerin, the gRNA (CGCCCACGCCCG AGTCCGCC) was also inserted in the same vector as for FLYWCH1. Then, Emerin homology arms were inserted in both sites of BSD-P2A-Turbo. Homology arms were amplified from cDNA using the primers HA1emerinMluI-Fw, HA1emerinNcoI-Rv, HA2emerinHindIII-Fw, and HA2emerinAscI-Rv. For both FLYWCH1 and Emerin, the length used for the homology arms was ~500 bp upstream and downstream from the start transcription site. The two homology arm vectors were transfected as a ratio 1:1 in HeLa cells and then the tagged cells were selected with blasticidin.

**Protein extract and streptavidin pulldown of Turbo and miniturbo-tagged cell lines**. WT and tagged cells were treated with 50 mM biotin (B20656, Life technologies) in a culture medium for one hour at 37 °C. The cells were washed with PBS and collected by scraping. Cells were then pelleted and nuclei were isolated using Nuclear Isolation Buffer (15 mM Tris pH 7.5, 15 mM NaCl, 60 mM KCl, 5 mM MgCl₂, 1 mM CaCl₂, 250 mM Sucrose and 0.03% NP40). Cells were incubated in this buffer for 30 min in a rotation wheel at 4 °C and then centrifuged at $3200 \times g$ for 10 min at 4 °C. Isolated nuclei were resuspended in RIPA buffer and sonicated in a Bioruptor until the extract was almost transparent. Samples were then centrifuged at maximum speed for 10 min at 4 °C to eliminate debris. The supernatant was collected and protein concentration was measured using Bradford assay. Approximately 4 mg of protein per reaction was incubated with 12.5 μl Streptavidin Sepharose High-Performance beads (15511301, Cytiva) and 50 μg/ml (final concentration) of Ethidium bromide (Sigma) for 2 h in a rotation wheel at 4 °C. Finally, samples were processed for LC–MS analyses using the same procedure as described for the ProtA-Turbo protocols.

**Western blotting**. Samples were incubated in an SDS loading buffer for 5 min at 95 °C after which proteins were separated on an SDS–PAGE gel. Proteins were transferred from the gel to a nitrocellulose membrane using a Trans-Blot Turbo Transfer System (Bio-Rad) according to the manufacturer's instructions. Membranes were blocked in blocking solution (5% milk in 0.1% Tween-PBS or 5% BSA in 0.1% Tween-PBS for the membranes that were used to stain biotin) in a shaker for 30 min at RT. Membranes were incubated in a shaker overnight at 4 °C with the antibody diluted in blocking solution, apart from V5 antibody, which was incubated in PBS + 0.1% v/v Tween. The antibodies used in this study were α-actin

(A1978, SIGMA) diluted 1:5000, α-BRG1 (Bethyl, A300-813A) diluted 1:1000, α-H3K9me3 (Abcam, ab8898) diluted 1:1000, α-emerin (10351-1-AP, Proteintech) diluted 1:1000, α-V5 antibody (P/N 46-0705, Invitrogen) diluted 1:1000, α-FLAG (Sigma, F3165) diluted 1:1000, α-SENP1 (Novus Biologicals, NBP2-55420) diluted 1:1000, α-SENP7 (Bethyl laboratories, A302-995A-T) diluted 1:1000 and α-HP1α (Cell Signaling, 2616S) diluted 1:1000. Next, membranes were incubated with HRP secondary antibody (Polyclonal Rabbit anti-Mouse Immunoglobulins/HRP P0260 or Polyclonal Swine anti-Rabbit Immunoglobulins/HRP P0399, both of them from Agilent Dako) diluted 1:1000 in PBS with 0.1% v/v Tween. Membranes blotted for biotin were incubated with HRP-Streptavidin (5911, Invitrogen) diluted 1:1000 in blocking solution with BSA for 1 h at room temperature. All of the membranes were developed using Supersignal West Pico Plus Chemiluminescent Substrate reagent (34580, Thermo Scientific) and imaged using ImageQuant LAS4000.

**Nuclear protein extraction**. Cells were resuspended in the hypotonic Buffer A (10 mM HEPES pH 7.9, 1.5 mM MgCl₂, 10 mM KCl, 0.1% NP40, 0.5 mM DTT), incubated 10 min on ice and centrifuged at $400 \times g$ for 5 min at 4 °C. The pellet was resuspended again with Buffer A containing 0.15% v/v NP40 and transferred to a dounce homogenizer (type B pestle) to mechanically break the cytoplasmic membranes. The sample was then centrifuged at $3200 \times g$ for 15 min at 4 °C. The nuclear pellet was washed with PBS once without resuspending the nuclei. Then, nuclei were incubated with Buffer C (420 mM NaCl, 20 mM HEPES pH 7.9, 20% glycerol, 2 mM MgCl₂, 0.2 mM EDTA, 0.1% NP40, 0.5 mM DTT) in a rotation wheel for 1 h at 4 °C and centrifuged at $21,000 \times g$ for 30 min at 4 °C. The supernatant represents the nuclear extract[47,48].

**Peptide pulldown**. The peptides used in this experiment contain the first 17 amino acids from histone H3 with or without trimethylation on lysine 9 (H3K9me3) followed by two glycines and a biotinylated lysine and they were synthesized using the Fmoc strategy[49,50]. Peptides were dissolved in Buffer A (150 mM NaCl, 50 mM Tris pH 8, 0.1% NP40) and incubated with 75 μl of Dynabeads MyOne Streptavidin C1 (Invitrogen, 65001) previously equilibrated with Buffer A. Beads were then washed twice with Buffer A and incubated with 500 μg nuclear extract diluted in dilution buffer (150 mM NaCl, 50 mM Tris pH 8, 2% NP40, 0.5 mM DTT, 10 μM ZnCl₂) in a rotation wheel for 3 h at 4 °C. After incubation, beads were washed three times with 1 ml dilution buffer containing 350 mM NaCl and twice with 1 ml dilution buffer. 40 μl of SDS loading protein buffer was then added to the beads after which samples were incubated for 5 min at 95 °C and loaded on an SDS–PAGE gel for western blot analyses.

**DNA pulldown**. The sequences of the oligonucleotides used for the DNA affinity purifications can be found in Supplementary Table 1. Oligos were annealed in Annealing buffer (20 mM Tris pH 8, 100 mM NaCl, 2 mM EDTA, forward oligo 20 μM, reverse oligo 30 μM) by heating the mixture at 95 °C for 10 min and then cooling it down to RT slowly. 20 μl Streptavidin Sepharose High-Performance beads (15511301, Cytiva) were washed with PBS + 0.1% NP40 and DNA binding buffer (10 mM Tris pH 8, 1 M NaCl, 1 mM EDTA, 0.05% NP40). Beads and oligos were incubated for 30 min in a rotation wheel at 4 °C. Beads were then washed once with DNA-binding buffer and twice with protein incubation buffer (50 mM Tris pH 8, 150 mM NaCl, 0.5% NP40, 1 mM DTT). Beads were then incubated with 500 μg of nuclear extracts together with DNA competitors (5 μg polydIdC and 5 μg polydAdT) for 90 min in a rotation wheel at 4 °C. Beads were washed three times with protein incubation buffer and twice with PBS. Then, 40 μl protein loading buffer was added to the beads and the mixture was incubated 10 min at 95 °C. Finally, samples were loaded on an SDS–PAGE gel for Western blotting.

**Reporting summary**. Further information on research design is available in the Nature Research Reporting Summary linked to this article.

## Data availability
The mass spectrometry proteomics data have been deposited to the ProteomeXchange Consortium via the PRIDE[51] partner repository with the dataset identifier PXD025012 and processed data are provided in the Source Data file. ChIP-sequencing data can be found under the reference number GSE169317 in the GEO database. Reference H3K9me3 ChIP-seq was downloaded from the GEO database under reference number GSE86814. Source data are provided with this paper.

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

## Acknowledgements

We thank members of the Vermeulen lab for fruitful discussions. The Vermeulen lab is part of the Oncode Institute, which is partly funded by the Dutch Cancer Society. Furthermore, this work is supported by an ERC Consolidator Grant to M.V. (771059). I.S.-B. is supported by a Marie Sklodowska-Curie post-doc fellowship under the European Union's Horizon 2020 research and innovation program (grant number 835908), G.v.M. is supported by a FEBS long term fellowship. We thank Lieke Lamers for preparing the libraries for sequencing and Pascal Jansen and Marijke Baltissen for technical support. We thank Silke Lochs and Jop Kind for sharing the pUC57 modified plasmid. Furthermore, we thank Dr. Thomas Jenuwein for providing the Suv39h1/2 double knock-out and wild-type mouse ESCs.

## Author contributions

I.S.-B., G.v.M., and M.V. conceived the project. I.S.-B. and G.v.M. performed experiments and analyses. I.S.-B., G.v.M., and M.V. interpreted the data and wrote the manuscript.

## Competing interests

The authors declare no competing interests.
