## [Peer Review File · Nature Communications]

REVIEWER COMMENTS

Reviewer #1 (Remarks to the Author):

The manuscript “‘Off the shelf’ proximity biotinylation for interaction proteomics” by Barriopedro et al. nicely introduces a novel method to perform interaction proteomics using biotinylation. The authors fused the TurboID proximity biotinylation enzyme to Protein A, introduced it into cells and by using a bait-specific antibody, interacting proteins of this protein or modification of interest are proximity biotinylated and can be identified via mass spectrometry (MS). By using antibodies against Emerin, H3K9me3 and BRG1 the authors nicely and convincingly demonstrate the feasibility of their approach and further identified FLYWCH1 as novel H3K9me3-interacting protein in centromeric heterochromatin. This study is well controlled and executed and I have only few comments.

My major and minor concerns are listed in detail below:

Major concerns:

- The authors state that all ProtA-turbo experiments followed by streptavidin-based affinity enrichments were performed in triplicates. Yet only one experiment is shown as scatter plot. In order to judge reproducibility Venn diagrams as well as heat maps showing interactors of all three experiments should be depicted.
- At least some of the identified novel interactors should be confirmed by regular immunoblotting without using biotinylation.
- The discovery of FLYWCH1 as H3K9me3 binding protein is exciting and raises several biological questions: Is FLYWCH1 a direct H3K9me3 binder (it should be in close proximity based on the assay conditions)? And if so, why was it not identified in other assays, such as H3K9me3 peptide pull-down approaches (see e.g. Eberl HC et al., Mol Cell, 2013)? The authors should use Suv39h double knock-out (DKO) mouse embryonic fibroblast (MEF) cells (Peters AH et al., Cell, 2001) that lost H3K9me3 completely and repeat their ProtA-turbo and ChIP-seq experiments. Further, H3K9me3 peptide-pull down experiments followed by FLYWCH1 immunoblots should be performed.

Minor:

- Immunofluorescence microscopy pictures containing one enlarged shot of one cells should indicate this particular cell in the composite record (e.g. Fig.1b, 2b, etc). Also, the changed scale bar should be indicated.
- Biotin ChIP-seq after H3K9me3 targeting (Fig. 1e,f) shows high background when compared to IgG control. Is this only due to the large heterochromatin loci H3K9me3 resides in? Why did the authors choose an H3K9me3 antibody in the first place?

- Is FLYWCH1 still bound to chromatin during mitosis? If so, chromosome spreads and co-staining with FLYWCH1 and CENP-A antibodies should reveal centromeric enrichment.

- In several Volcano plots many “background” proteins depicted in light gray are observed. What are these proteins and why were these not depicted?

- Why was FLYWCH1 tagged with a miniTurbo enzyme that is less active than the TurboID enzyme and why was not the ProtA-turbo experiment with either an anti-GFP, -V5 or -FLYWCH1 antibody performed?

Reviewer #2 (Remarks to the Author):

This work describes interesting approach for mapping interactome of specific nuclear protein including modified histone protein using antibody-based TurboID. Although this work showed interesting results and useful technique for biological community, I found that current manuscript requires revision and addition for the possible publication in Nature Communications.

1. I could understand the reason using antibody-based proximity labeling approach for mapping the biomolecules that interact with post-translationally modified histone because it should be complicated to map it with exogenous protein expression approach. However, the introduction of antibody-conjugated TurboID should require fixation or permeabilization steps that can usually compromise ultrastructure. In the electron microscope imaging experiments, it is well recognized that permeabilization step may disturb the ultrastructures of protein complexes and many of researchers prefer to image the fixed sample without permeabilization and the same issue should be in this method. Thus, the authors should clearly make a cautionary note that these procedures can affect protein composition and structures of subcellular complexes in the Discussion part of this manuscript.

2. Among the ProtA-TurboID's identified proteins, FLYWCH1 was revealed to be colocalized with H3K9me3 at the(peri)centromeric heterochromatin region by follow-up studies. However, it is still uncertain whether other selected proteins in the interactome list of H3K9me3 (Supp Dataset 1) can also have colocalization with H3K9me3. The authors should show more followup results that can confirm their proteomic results.

3. Among the identified proteins as H3K9me3 interactome (Fig. 2d, Supp Dataset 1), HELLS was enriched in both fixed and unfixed samples and it showed the higher enrichment value than FLYWCH1, indicating that HELLS might be a strong candidate of H3K9me3-interacting protein but the authors mentioned it as common streptavidin contaminants in the Figure and Figure legends (i.e. Fig. 2 and Fig. 3) and I could not find the detail information about this "common streptavidin contaminants" in the main text. If HELLS is a common streptavidin contaminant, it should be also detected in the Streptavidin-enriched sample of control sample (i.e. α -IgG) and HELLS should not be shown in enriched proteins of H3K9me3 in volcano plot analysis. However, data tells that HELLS is specifically enriched in the anti-H3K9me3:ProtA-TurboID sample and it means HELLS is highly likely not a "common" streptavidin contaminant. Thus, the authors should provide clear explanation why they exclude this protein (and others) from their followup studies and it is required to confirm they whether HELLS can be a co-localized interacting protein with H3K9me3 protein because their proteomic data strongly indicates it.

Reviewer #3 (Remarks to the Author):

In the manuscript "‘Off the shelf’ proximity biotinylation for interaction proteomics" the authors present a new approach to proximity labeling using a ProtA-Turbo enzyme fusion for interaction proteomics that does not require genetic manipulation of cells and has significant ease of use advantages. Overall, this is a valuable approach and well presented. There are some minor corrections and clarifications required. Most notably the sparse and unclear description of statistical parameters and statistical methods needs to be corrected before publication.

One key limitation that should be discussed is that antibodies, while very useful, have their own limitations that are inherited by this approach. These include issues like epitope occlusion and limited specificity relative to the dynamic range of protein abundances. While the validation data here is strong and does not appear to suffer from these issues, the creation of a Turbo-target protein fusion, while not perfect either, will not suffer from these particular issues.

It would also be useful to comment on the differences between the fixed and non-fixed experiments. Particularly, Emerin results have very different fold changes. It is much larger in the fixed. The H3K9me3 fold change shows the opposite differences between the experiments. The unfixed H3K9me3 exhibits a much greater fold change than its fixed counterpart. They both basically recapitulate each other but behave differently in how much enrichment is achieved but is not consistent from protein to protein. Is this fundamental to the protein/PTM target and reveals some biases between the experiments? How much of this is just regular experimental variability?

The description of statistical methods is lacking and it is difficult to determine precisely what the procedures are. “Cells are then lysed using high stringency lysis and biotinylated proteins are affinity enriched in triplicate” – This is unclear if the triplicates originate in fully independent experiments (i.e. starting with cells), or if aliquots of the same cells are affinity enriched. “The triplicates were grouped based on experimental condition and only proteins that were reproducibly quantified in one of the triplicates were maintained for downstream analyses.” – Again unclear if these are biological or technical triplicates. It is also unclear what reproducibly quantified means. This infers that there are multiple measurements in each replicated (and thus reproducibly quantified). How many technical replicates then? What are your criteria for reproducibly quantified? Simple observation (non-zero quantity) in all (or 2/3) technical replicates? “Statistically different proteins were identified using a t-test (FDR < 0.05) and required additionally at least a 1.5-fold...” – Is this a Welch’s two-tailed T-test? I recommend a paragraph describing these details in one place in materials and methods but also greater clarity in the main text.

p2 Line 51

You should cite the original work of each of these techniques here.

p3 line 88

“for in vivo interaction proteomics” some would disagree with your use of in vivo. ex vivo, or in cell culture might be more appropriate.

line 429

“C18 Stagetips” should be specified as to manufacturer. I assume you are using 3M Empore material?

Figure 2D

The order of the K9me3 volcano plot is inconsistent with Figure 2C and Figure 1D.

Best,

Nicolas L. Young, Ph.D.

Baylor College of Medicine

Houston, TX, USA

We thank the reviewers for their positive and constructive comments on our manuscript. Based on this and additional comments from the editor, we have performed numerous experiments to strengthen our manuscript. These experiments are detailed below in the point by point response and they are also summarized in the cover letter. Our answers to the reviewer comments are marked in bold. Textual changes in the manuscript are marked in yellow.

Reviewer #1 (Remarks to the Author):

The manuscript “‘Off the shelf’ proximity biotinylation for interaction proteomics” by Barriopedro et al. nicely introduces a novel method to perform interaction proteomics using biotinylation. The authors fused the TurboID proximity biotinylation enzyme to Protein A, introduced it into cells and by using a bait-specific antibody, interacting proteins of this protein or modification of interest are proximity biotinylated and can be identified via mass spectrometry (MS). By using antibodies against Emerin, H3K9me3 and BRG1 the authors nicely and convincingly demonstrate the feasibility of their approach and further identified FLYWCH1 as novel H3K9me3-interacting protein in centromeric heterochromatin. This study is well controlled and executed and I have only few comments. My major and minor concerns are listed in detail below:

We thank the reviewer for his/her kind words about our manuscript.

Major concerns:

- The authors state that all ProtA-turbo experiments followed by streptavidin-based affinity enrichments were performed in triplicates. Yet only one experiment is shown as scatter plot. In order to judge reproducibility Venn diagrams as well as heat maps showing interactors of all three experiments should be depicted.

As the reviewer correctly pointed out, affinity enrichments were performed in triplicate, both for the specific antibody and for the negative IgG control. Thus, for each volcano plot shown in the manuscript, three specific and three control affinity purifications were performed, not just one. Performing triplicates allows performing a t-test and facilitates permutation-based FDR filtering. Proteins specifically enriched in the experiment compared to the negative control have to be consistently enriched in the three replicates in order to receive a low p-value and these proteins end up on top of the volcano plot (Y axis displays $-\text{Log}_{10}$ transformed p-value). Thus, even though there is just one dot visualized per protein, this dot is the result of comparing three replicates for the IgG control with three replicates of the specific antibody. To further illustrate the technical reproducibility of our experiments, we also visualized some of our data in a clustering (figure S1e). This clearly shows that bait-proximal hits identified in various ProtA-Turbo experiments are reproducibly enriched in three specific affinity enrichments compared to the negative IgG control.

- At least some of the identified novel interactors should be confirmed by regular immunoblotting without using biotinylation.

Many of the novel interactors we identified for various baits (i.e Flywch1 for H3K9me3) are indirect interactors for these baits. These proteins are in vivo proximal but are not directly interacting with the bait. Thus, these proteins can only be identified/validated using the proximity biotinylation approach. However, we have confirmed some novel identified bait-proximal proteins using immunoblotting with antibodies against these novel interactors in specific H3K9me3 affinity enrichments versus IgG controls (added as Figure S3b-c) in the revised manuscript. We have also confirmed that FLYWCH1 is not a direct H3K9me3 interactor using a peptide pull-down approach, see also the point below (Fig. S3i of the revised manuscript).

- The discovery of FLYWCH1 as H3K9me3 binding protein is exciting and raises several biological

questions: Is FLYWCH1 a direct H3K9me3 binder (it should be in close proximity based on the assay conditions)? And if so, why was it not identified in other assays, such as H3K9me3 peptide pull-down approaches (see e.g. Eberl HC et al., Mol Cell, 2013)? The authors should use Suv39h double knock-out (DKO) mouse embryonic fibroblast (MEF) cells (Peters AH et al., Cell, 2001) that lost H3K9me3 completely and repeat their ProtA-turbo and ChIP-seq experiments. Further, H3K9me3 peptide-pull down experiments followed by FLYWCH1 immunoblots should be performed.

To address this point, as suggested by the reviewer, we performed an H3K9me3 peptide pull-down experiment combined with western blotting for FLYWCH1 (Fig. S3i in the revised manuscript). This experiment revealed that FLYWCH1 is not a direct reader protein for H3K9me3 but instead an indirect H3K9me3 proximal protein in vivo, which is in agreement with previous peptide pull-down based studies such as the one mentioned by the reviewer, in which FLYWCH1 was indeed not identified as a H3K9me3 reader. We also performed DNA pull-down experiments using simple repeat sequences and predicted FLYWCH1-motifs as a bait, which revealed specific enrichment of FLYWCH1 (Fig. S3j in the revised manuscript). This suggests that FLYWCH1 binds repeat DNA directly, which is very plausible given the 5 Flwch1 zinc fingers the protein contains, which are predicted DNA binding domains. These simple repeat regions in the genome are strongly enriched for H3K9me3, which explains why FLYWCH1 was identified in H3K9me3 ProtA-Turbo experiments. We also performed H3K9me3 ProtA-Turbo experiments in wild-type mouse embryonic stem cells versus Suv39h1/2 knock-out embryonic stem cells. As expected, these experiments revealed a reduction of various H3K9me3 readers compared to WT cells, along with a strong reduction in Suv39h1/2 itself and a range of proteins associated with (peri)centromeric, H3K9me3 marked heterochromatin regions (fig. S2h-i in the revised manuscript). Of note, Suv39h1/2 double knock-out cells still contain significant amounts of H3K9me3, catalyzed by SetDB1/2 and EHMT1/2. A complete loss of H3K9me3 proximal proteins in WT versus Suv39h1/2 knock-out cells is therefore not to be expected. This experiment further illustrates the broad applicability of the ProtA-Turbo enzyme in embryonic stem cells and in combination with perturbation experiments, in this case WT versus knock-out cells. We also note that in these ProtA-Turbo H3K9me3 targeting experiments in mouse ESCs, we did not detect a single FLYWCH1 peptide. This is due to the fact that FLYWCH1 is low expressed in ESCs, even in deep whole proteome analyses in mouse ESCs, Flywch1 is not detected (for example in Yang, Cell Systems, 2019; PMID: 31078527).

Minor:

- Immunofluorescence microscopy pictures containing one enlarged shot of one cells should indicate this particular cell in the composite record (e.g. Fig.1b, 2b, etc). Also, the changed scale bar should be indicated.

This has been corrected in the revised manuscript.

- Biotin ChIP-seq after H3K9me3 targeting (Fig. 1e,f) shows high background when compared to IgG control. Is this only due to the large heterochromatin loci H3K9me3 resides in? Why did the authors choose an H3K9me3 antibody in the first place?

The biotin ChIP-seq signal is always expected to be more noisy compared to a classical ChIP-seq experiment since the ProtA-turbo enzyme is biotinylating H3K9me3-proximal proteins within a ~10 nM radius. A biotin ChIP-seq will therefore show more background compared to a classical ChIP-seq track. This is for example also visible in biotin ChIP-seq after the targeting histone marks using reader domains as done in ChromID (Villasenor, Nat. Biotech, 2020; PMID: 32123383). H3K9me3 was chosen to serve as a proof of principle that histone modifications can be targeted by the ProtA-turbo enzyme. Furthermore, the commercial H3K9me3 antibody is very specific and of high quality. Finally, many direct and indirect H3K9me3 proximal proteins have previously been

described, this modification thus serves as an excellent bait to benchmark the ProtA-Turbo enzyme.

- Is FLYWCH1 still bound to chromatin during mitosis? If so, chromosome spreads and co-staining with FLYWCH1 and CENP-A antibodies should reveal centromeric enrichment.

To address this point, we performed microscopy experiments in metaphase-arrested cells, which showed that FLYWCH1 remains bound to chromatin across the cell cycle. Furthermore, FLYWCH1 and INCENP co-localize at a subset of centromeres, thus confirming our FLYWCH1 ChIP-seq results, which revealed binding to a subset of (peri)centromeres in human cells (Fig. S3e).

- In several Volcano plots many “background” proteins depicted in light gray are observed. What are these proteins and why were these not depicted?

If the reviewer indicates the light grey names, with these we indicated common streptavidin contaminants as listed in the so-called CRAPome repository (Mellacheruvu, Nat. Methods, 2013; PMID 23921808), which is why we marked these in light grey to alert the reader that these are most likely false positive hits. To prevent confusion, we have now removed these names (as also further discussed in a point of reviewer 2). If the reviewer refers to additional grey dots without a name, these comprise novel candidate proximal proteins of the targeted protein. To prevent overcrowding of the plot, we left these names out. However, the data underlying these plots accompanies the manuscript as Table S2, which will allow readers to browse through the data and see which proteins are found as enriched in the baits compared to the IgG control.

- Why was FLYWCH1 tagged with a miniTurbo enzyme that is less active than the TurboID enzyme and why was not the ProtA-turbo experiment with either an anti-GFP, -V5 or -FLYWCH1 antibody performed?

We tried to generate a Turbo-FLYWCH1 knock-in line, but this failed. We did, however, manage to generate a miniTurbo-FLYWCH1 knock-in line, which we then used in the study to identify the Flywch1 in vivo proximal proteome. Performing direct proximity biotinylation with this cell line seemed more appropriate compared to targeting Flywch1 using ProtA-turbo using a V5 or GFP antibody. But this latter approach could in principle also be pursued in future experiments. We now also comment on this in the manuscript.

Reviewer #2 (Remarks to the Author):

This work describes interesting approach for mapping interactome of specific nuclear protein including modified histone protein using antibody-based TurboID. Although this work showed interesting results and useful technique for biological community, I found that current manuscript requires revision and addition for the possible publication in Nature Communications.

We thank the reviewer for his/her kind words about our work.

1. I could understand the reason using antibody-based proximity labeling approach for mapping the biomolecules that interact with post-translationally modified histone because it should be complicated to map it with exogenous protein expression approach. However, the introduction of antibody-conjugated TurboID should require fixation or permeabilization steps that can usually compromise ultrastructure. In the electron microscope imaging experiments, it is well recognized that permeabilization step may disturb the ultrastructures of protein complexes and many of

researchers prefer to image the fixed sample without permeabilization and the same issue should be in this method. Thus, the authors should clearly make a cautionary note that these procedures can affect protein composition and structures of subcellular complexes in the Discussion part of this manuscript.

This is a valid point and we included a cautionary note in the discussion concerning the fact that permeabilisation and fixation may affect subcellular structures, as indicated by the reviewer.

2. Among the ProtA-TurboID's identified proteins, FLYWCH1 was revealed to be colocalized with H3K9me3 at the(peri)centromeric heterochromatin region by follow-up studies. However, it is still uncertain whether other selected proteins in the interactome list of H3K9me3 (Supp Dataset 1) can also have colocalization with H3K9me3. The authors should show more followup results that can confirm their proteomic results.

To address this point, we performed additional immunofluorescence and H3K9me3 ProtA-turbo targeting experiments followed by western blotting for three novel H3K9me3 interactors: Rnf169, Senp1 and Senp7 (included as Fig. S3b and S3c in the revised manuscript)

3. Among the identified proteins as H3K9me3 interactome (Fig. 2d, Supp Dataset 1), HELLS was enriched in both fixed and unfixed samples and it showed the higher enrichment value than FLYWCH1, indicating that HELLS might be a strong candidate of H3K9me3-interacting protein but the authors mentioned it as common streptavidin contaminants in the Figure and Figure legends (i.e. Fig. 2 and Fig. 3) and I could not find the detail information about this "common streptavidin contaminants" in the main text. If HELLS is a common streptavidin contaminant, it should be also detected in the Streptavidin-enriched sample of control sample (i.e. α -IgG) and HELLS should not be shown in enriched proteins of H3K9me3 in volcano plot analysis. However, data tells that HELLS is specifically enriched in the anti-H3K9me3:ProtA-TurboID sample and it means HELLS is highly likely not a "common" streptavidin contaminant. Thus, the authors should provide clear explanation why they exclude this protein (and others) from their followup studies and it is required to confirm they whether HELLS can be a co-localized interacting protein with H3K9me3 protein because their proteomic data strongly indicates it.

HELLS is included in the so-called CRAPome database (Mellacheruvu, Nat. Methods, 2013; PMID 23921808), which lists common contaminants (false positives) in various affinity purification experiments, including streptavidin-based affinity purifications. To alert the readers, we therefore decided to label all proteins in our plots, which are commonly observed as hits in streptavidin-based affinity purifications. For example, HELLS was identified as a streptavidin interactor in 97 out of 716 streptavidin based affinity enrichment experiments with a range of baits included in the CRAPome repository. It could, however, still be possible that HELLS is a real H3K9me3 proximal protein inside cells, but further orthogonal validation experiments are needed for his. To address the reviewer's comment and prevent future confusion for the reader, we have removed the name HELLS from the plot.

Reviewer #3 (Remarks to the Author):

In the manuscript "Off the shelf' proximity biotinylation for interaction proteomics" the authors present a new approach to proximity labeling using a ProtA-Turbo enzyme fusion for interaction proteomics that does not require genetic manipulation of cells and has significant ease of use advantages. Overall, this is a valuable approach and well presented. There are some minor corrections and clarifications required. Most notably the sparse and unclear description of statistical parameters and statistical methods needs to be corrected before publication.

We thank the reviewer for his kind words.

One key limitation that should be discussed is that antibodies, while very useful, have their own limitations that are inherited by this approach. These include issues like epitope occlusion and limited specificity relative to the dynamic range of protein abundances. While the validation data here is strong and does not appear to suffer from these issues, the creation of a Turbo-target protein fusion, while not perfect either, will not suffer from these particular issues.

This is a valid point and we included cautionary comments along these lines in the discussion of our revised manuscript.

It would also be useful to comment on the differences between the fixed and non-fixed experiments. Particularly, Emerin results have very different fold changes. It is much larger in the fixed. The H3K9me3 fold change shows the opposite differences between the experiments. The unfixed H3K9me3 exhibits a much greater fold change than its fixed counterpart. They both basically recapitulate each other but behave differently in how much enrichment is achieved but is not consistent from protein to protein. Is this fundamental to the protein/PTM target and reveals some biases between the experiments? How much of this is just regular experimental variability?

The reviewer is correct when stating that the specific interactions we detect for different baits are quite similar using the fixed and non-fixed workflow, which is reassuring. However, as the reviewer correctly states, we indeed observe some differences concerning the enrichment values that are obtained in various experiments. We believe this is of technical nature and related to small differences in washing conditions, generated lysates, etc, but not related, for example, to differences between using antibodies targeting PTMs or proteins. These experiments are not yet automated but performed by hand, so small technical variabilities between experiments are difficult to avoid. But as mentioned before, it is very reassuring to observe an overlapping set of proteins as statistically significant interactors in both the fixed and non-fixed workflow. We added a sentence about these technical variations in the discussion.

The description of statistical methods is lacking and it is difficult to determine precisely what the procedures are. "Cells are then lysed using high stringency lysis and biotinylated proteins are affinity enriched in triplicate" – This is unclear if the triplicates originate in fully independent experiments (i.e. starting with cells), or if aliquots of the same cells are affinity enriched. "The triplicates were grouped based on experimental condition and only proteins that were reproducibly quantified in one of the triplicates were maintained for downstream analyses." – Again unclear if these are biological or technical triplicates. It is also unclear what reproducibly quantified means. This infers that there are multiple measurements in each replicated (and thus reproducibly quantified). How many technical replicates then? What are your criteria for reproducibly quantified? Simple observation (non-zero quantity) in all (or 2/3) technical replicates? "Statistically different proteins were identified using a t-test (FDR < 0.05) and required additionally at least a 1.5-fold..." – Is this a Welch's two-tailed T-test? I recommend a paragraph describing these details in one place in materials and methods but also greater clarity in the main text.

We thank the reviewer for pointing out these shortcomings concerning the experimental and statistical descriptions of our experiments. We edited our manuscript accordingly and expanded the material and methods section concerning the affinity purifications and statistical filtering of the interaction proteomics data.

p2 Line 51

You should cite the original work of each of these techniques here.

Corrected.

p3 line 88

“for in vivo interaction proteomics” some would disagree with your use of in vivo. ex vivo, or in cell culture might be more appropriate.

Corrected.

line 429

“C18 Stagetips” should be specified as to manufacturer. I assume you are using 3M Empore material?

Indeed, this refers to 3M Empore material. This has been corrected.

Figure 2D

The order of the K9me3 volcano plot is inconsistent with Figure 2C and Figure 1D.

Corrected.

Best,

Nicolas L. Young, Ph.D.
Baylor College of Medicine
Houston, TX, USA

REVIEWERS' COMMENTS

Reviewer #1 (Remarks to the Author):

The authors have addressed all my comments and suggestions to my satisfaction. I recommend publication of the manuscript.

Reviewer #2 (Remarks to the Author):

It is my pleasure to see the improved manuscript after the revision process. In the revised manuscript, I could see the addition of cautionary notes in the discussion (related to my comment #1) and additional experimental results of Figure S3b and S3c (related to my comment #2) look great. Related to my comment #3, the authors decided to remove "common streptavidin contaminant protein" mark on the HELLS in the plot, however, these marks or notes on the HELLS and other protein(s) which are overlapped with the CRAPome (Mellacheruvu et al. 2013) are important because these notes will lead the readers to focus on the real findings. Thus, I recommend to mark the overlap with CRAPome data in the plot again and also mark in the spreadsheet too. Since this issue is caused by non-specific binding affinity of non-biotinylated protein on the bead surface, it might be better to note that this issue can be clearly addressed if the biotin-modified peptides can be analysed by mass spectrometer as shown in the previous studies (Refs: Lee SY et al. ACS Central Science, 2016, 2, 506-516, PMID: 27610411; Kwak C et al. PNAS, 2020, 117, 12109-12120, PMID: 32414919; Lee SY et al. Methods Mol Biol, 2019, 97-105, PMID: 31124091). After this minor revision process, I believe that this manuscript would be worth publishing in Nature Communications.

Reviewer #3 (Remarks to the Author):

In the manuscript "'Off the shelf' proximity biotinylation for interaction proteomics" the authors present a new approach to proximity labeling using a ProtA-Turbo enzyme fusion for interaction proteomics that does not require genetic manipulation of cells and has ease of use advantages. Overall, this is a valuable approach and well presented. The authors have been effective in the revision and improvement of the manuscript, including additional experiments. There remains one point that was not quite effectively communicated or understood and very minor changes to the text are recommended.

“However, all widely applicable techniques that require the use of antibodies, such as CHIP-seq, suffer from the same limitations.”

- This sentence makes a valid but not particularly relevant comparison. The more salient point is that many other versions of proximity labeling do not use antibodies.

This difference should be noted and the value and limitations of each should be concisely noted. A fusion protein has its own limitations and potential for artifactual results. It is no longer the native protein. There are costs and benefits to both approaches. This comparison should be the focus of the discussion. I would further argue that there is a benefit to the development and use of complementary methods with different limitations. You do an excellent job of comparing and contrasting to other approaches on other aspects. The antibody-based approach opens some opportunities that are not available by other approaches. E.g. targeting a post-translational modification, as done with H3K9me3. Furthermore, it is more modular and easier to implement quickly for many targets. However, it indeed brings along all of the attributes of antibodies, including the limitations that are present in all antibody-based methods.

Nicolas Young

Baylor College of Medicine

Houston, TX

Reviewer #1 (Remarks to the Author):

The authors have addressed all my comments and suggestions to my satisfaction. I recommend publication of the manuscript.

Reviewer #2 (Remarks to the Author):

It is my pleasure to see the improved manuscript after the revision process. In the revised manuscript, I could see the addition of cautionary notes in the discussion (related to my comment #1) and additional experimental results of Figure S3b and S3c (related to my comment #2) look great. Related to my comment #3, the authors decided to remove "common streptavidin contaminant protein" mark on the HELLS in the plot, however, these marks or notes on the HELLS and other protein(s) which are overlapped with the CRAPome (Mellacheruvu et al. 2013) are important because these notes will lead the readers to focus on the real findings. Thus, I recommend to mark the overlap with CRAPome data in the plot again and also mark in the spreadsheet too. Since this issue is caused by non-specific binding affinity of non-biotinylated protein on the bead surface, it might be better to note that this issue can be clearly addressed if the biotin-modified peptides can be analysed by mass spectrometer as shown in the previous studies (Refs: Lee SY et al. ACS Central Science, 2016, 2, 506-516, PMID: 27610411; Kwak C et al. PNAS, 2020, 117, 12109-12120, PMID: 32414919; Lee SY et al. Methods Mol Biol, 2019, 97-105, PMID: 31124091). After this minor revision process, I believe that this manuscript would be worth publishing in Nature Communications.

We thank the reviewer for these suggestions. In the revised manuscript we have added to the Source Data a table that highlights how often proteins were identified in our ProtA-Turbo experiments targeting Emerin, BRG1 and H3K9me3 (fixed and unfixed). This mainly shows that HELLS (found in 5/6 experiments) and IFI16 (found in 6/6 experiments) were detected with very high frequencies independent of the bait targeted (indicating potential a specific binding). We have thus marked these proteins in grey in the volcano plot and highlighted in the Figure legends that these proteins might comprise contaminants. In addition, to the Source Data we have added an overview table that shows for every protein that was enriched in one of the ProtA-Turbo experiments shown in Figure 1d and 2d (relative to IgG) how often these are present in experiments listed in the Crapome database. In the methods we now also describe how to interpret these values. In this way the reader can objectively judge whether a protein of interest might comprise a contaminant and thus requires orthogonal validation for localization near the bait protein of interest.

Reviewer #3 (Remarks to the Author):

In the manuscript "Off the shelf proximity biotinylation for interaction proteomics" the authors present a new approach to proximity labeling using a ProtA-Turbo enzyme fusion for interaction proteomics that does not require genetic manipulation of cells and has ease of use advantages. Overall, this is a valuable approach and well presented. The authors have been effective in the revision and improvement of the manuscript, including additional experiments. There remains one point that was not quite effectively communicated or understood and very minor changes to the text are recommended.

285

"However, all widely applicable techniques that require the use of antibodies, such as ChIP-seq, suffer

from the same limitations.” - This sentence makes a valid but not particularly relevant comparison. The more salient point is that many other versions of proximity labelling do not use antibodies.

This difference should be noted and the value and limitations of each should be concisely noted. A fusion protein has its own limitations and potential for artifactual results. It is no longer the native protein. There are costs and benefits to both approaches. This comparison should be the focus of the discussion. I would further argue that there is a benefit to the development and use of complementary methods with different limitations. You do an excellent job of comparing and contrasting to other approaches on other aspects. The antibody-based approach opens some opportunities that are not available by other approaches. E.g. targeting a post-translational modification, as done with H3K9me3. Furthermore, it is more modular and easier to implement quickly for many targets. However, it indeed brings along all of the attributes of antibodies, including the limitations that are present in all antibody-based methods.

We thank the reviewer for explaining this valid point. We have now added a paragraph to the discussion that more clearly discusses this. We now write: “Both HRP and ProtA-Turbo based methods rely on the specificity of the used antibodies as well as the abundance of the proteins or modifications that are targeted. ProtA-Turbo targeting experiments on low abundant proteins or modifications may suffer from high background biotinylation signal since at any given time a number of ProtA-Turbo molecules may not be target engaged. ‘Classical’ proximity biotinylation experiments in which the bait of interest is directly fused to a biotinylation enzyme do not suffer from these drawbacks. On the other hand, fusing of a proximity biotinylation enzyme to a bait protein of interest may negatively affect the function of that bait. Furthermore, PTMs are difficult to target using classical proximity biotinylation experiments but these can easily be targeted using the ProtA-Turbo enzyme. Thus, classical proximity biotinylation experiments and ProtA-Turbo mediated proximity biotinylation experiments each have their own pros and cons and are complementary in nature.”

Nicolas Young
Baylor College of Medicine
Houston, TX